# Histone H2A-H2B binding by Pol α in the eukaryotic replisome contributes to the maintenance of repressive chromatin

Cecile Evrin[1], Joseph D Maman[2], Aurora Diamante[2], Luca Pellegrini[2] & Karim Labib[1,*]

## Abstract

The eukaryotic replisome disassembles parental chromatin at DNA replication forks, but then plays a poorly understood role in the re-deposition of the displaced histone complexes onto nascent DNA. Here, we show that yeast DNA polymerase α contains a histone-binding motif that is conserved in human Pol α and is specific for histones H2A and H2B. Mutation of this motif in budding yeast cells does not affect DNA synthesis, but instead abrogates gene silencing at telomeres and mating-type loci. Similar phenotypes are produced not only by mutations that displace Pol α from the replisome, but also by mutation of the previously identified histone-binding motif in the CMG helicase subunit Mcm2, the human orthologue of which was shown to bind to histones H3 and H4. We show that chromatin-derived histone complexes can be bound simultaneously by Mcm2, Pol α and the histone chaperone FACT that is also a replisome component. These findings indicate that replisome assembly unites multiple histone-binding activities, which jointly process parental histones to help preserve silent chromatin during the process of chromosome duplication.

**Keywords** DNA polymerase alpha; DNA replication; histone chaperone; histones; replisome

**Subject Categories** Chromatin, Epigenetics, Genomics & Functional Genomics; DNA Replication, Repair & Recombination

The EMBO Journal (2018) 37: e99021

## Introduction

A subset of the many factors required to duplicate chromosomes assembles into a large and highly dynamic molecular machine called the replisome (Bell & Labib, 2016; Burgers & Kunkel, 2017; Kunkel & Burgers, 2017; Riera *et al*, 2017). Bacterial, viral and eukaryotic replisomes vary widely in their composition, but a common feature in all cases is that replisome assembly connects the DNA helicase that unwinds the parental DNA duplex to one or more of the DNA polymerases that synthesize the two daughter strands. Such physical connections between helicase and polymerases can be direct or else be mediated by other factors, and can serve to couple the rates of DNA unwinding and DNA synthesis, thus minimizing the exposure of single-strand DNA at replication forks. In addition, work with *Escherichia coli* showed that replisome assembly allows DNA polymerase action to propel the DNA helicase forwards and thus stimulate the rate of fork progression (Kim *et al*, 1996).

Whereas the *E. coli* replisome is well characterized and the role of each component is defined, the eukaryotic replisome remains an enigma and contains around twice as many subunits, many of which are still of unknown biochemical function. The greater complexity of the eukaryotic replisome reflects the additional challenges of duplicating chromosomes that contain vast amounts of DNA packaged with histones into chromatin, which must be disrupted and then re-established during replication, whilst preserving epigenetic information that controls patterns of gene expression (Hammond *et al*, 2017; Miller & Costa, 2017). Very little is known about the activities within the replisome that contributes to the processing of parental chromatin during chromosome duplication. Moreover, the links between parental histone processing at replication forks and the preservation of gene silencing are largely unexplored.

The eukaryotic replisome connects the CMG helicase (Cdc45-MCM-GINS) to two of the three DNA polymerases that are essential for DNA synthesis at eukaryotic replication forks, namely DNA polymerase epsilon (Pol ε) that synthesizes the leading strand and DNA polymerase alpha (Pol α) that initiates each new Okazaki fragment during lagging strand synthesis. Pol ε binds directly to the CMG helicase and stimulates the rate of fork progression (Kang *et al*, 2012; Sengupta *et al*, 2013; Georgescu *et al*, 2014; Langston *et al*, 2014), analogous to the stimulation of helicase by polymerase in the *E. coli* replisome (Kim *et al*, 1996). In contrast, Pol α is connected indirectly to the CMG helicase by the trimeric adaptor known as Ctf4 (Zhu *et al*, 2007; Gambus *et al*, 2009; Tanaka *et al*, 2009a), each protomer of which can bind to short "Ctf4-Interacting Peptides" (or CIP-boxes) in client proteins such as the Sld5 subunit of yeast CMG, or the Pol1 catalytic subunit of Pol α (Simon *et al*, 2014; Villa *et al*, 2016). The tethering of Pol α to the eukaryotic replisome by Ctf4 was originally thought to promote efficient priming of Okazaki

1 MRC Protein Phosphorylation and Ubiquitylation Unit, Sir James Black Centre, School of Life Sciences, University of Dundee, Dundee, UK
2 Department of Biochemistry, University of Cambridge, Cambridge, UK
   *Corresponding author. Tel: +44 1382 384108; E-mail: kpmlabib@dundee.ac.uk

fragments during lagging strand DNA synthesis. However, Ctf4 has no apparent impact on DNA synthesis *in vitro*, using a reconstituted replisome system based on purified budding yeast proteins (Yeeles *et al*, 2015, 2017). Moreover, recent work indicates that trimeric Ctf4 represents a hub in the eukaryotic replisome with roles that go beyond DNA synthesis (Fumasoni *et al*, 2015; Samora *et al*, 2016; Villa *et al*, 2016). In addition to Pol α, budding yeast Ctf4 recruits other CIP-box proteins such as the Chl1 helicase that helps to establish sister chromatid cohesion (Samora *et al*, 2016) and the Dna2 and Tof2 proteins that help to maintain the integrity of the rDNA repeats on chromosome 12 (Villa *et al*, 2016). Here, we show that replisome tethering of Pol α via Ctf4 is dispensable for efficient DNA synthesis in budding yeast cells, and instead is required to preserve epigenetic gene silencing at telomeres and the silent mating-type genes, in a manner that is dependent upon a novel histone-binding motif in the amino-terminal region of the Pol1 DNA polymerase subunit. These findings expand our view of the eukaryotic replication machinery and show how the coupling of a DNA polymerase to the helicase within the replisome can contribute to functions beyond DNA synthesis.

# Results

## Displacement of Pol α from Ctf4 does not perturb DNA replication *in vivo*

We previously showed that Pol α was no longer able to associate with Ctf4 in *pol1-4A* cells with mutations in the CIP-box motif of the catalytic subunit (Simon *et al*, 2014). However, *pol1-4A* cells lack multiple phenotypes of *ctf4Δ* strains, which instead reflect the recruitment by Ctf4 of other CIP-box proteins to the replisome (Samora *et al*, 2016; Villa *et al*, 2016). To examine in more detail the consequences of displacing Ctf4-tethered Pol α from the replisome, we synchronized control cells and *pol1-4A* in G1-phase and monitored DNA synthesis and replisome assembly when cells entered S-phase. As shown in Fig 1, the kinetics of DNA synthesis were very similar in *pol1-4A* and control cells (Fig 1A), and replisome assembly was normal except that Pol α was largely displaced (Fig 1B; Appendix Fig S1 shows that the weakened association of Pol1-4A with the replisome is equivalent to the situation previously reported for *ctf4Δ* cells (Sengupta *et al*, 2013), reflecting a residual Ctf4-independent link between Pol α and the CMG helicase).

As a more sensitive assay for impaired DNA synthesis, we combined *pol1-4A* with deletion of the *MEC1* gene, the yeast orthologue of the ATR checkpoint kinase, which becomes essential in cells that have the slightest defect in DNA replication. Colony growth of *mec1Δ* cells was unaffected by the *pol1-4A* mutations (Fig 1C), indicating that Pol α is still very efficient at priming nascent DNA during leading and lagging strand synthesis in *pol1-4A* cells, even though it is no longer tethered to the replisome by Ctf4. As a control, we confirmed that *mec1Δ* is synthetic lethal with the *pol1-F1463A* point mutation (Fig 1D), which displaces primase from the carboxyl terminus of Pol1 (Kilkenny *et al*, 2012). Previous work indicated that the progression of chromosome replication in *pol1-F1463A* is very similar to control cells, but cell viability is absolutely dependent upon Mec1, pointing to a subtle defect in DNA synthesis at replication forks (Kilkenny *et al*, 2012).

Overall, these findings indicated that replisome tethering of Pol α by Ctf4 is dispensable *in vivo* for efficient DNA synthesis and instead fulfils a different role during chromosome replication.

## Replisome tethering of Pol α is required to preserve gene silencing at sub-telomeric and mating-type loci

One possible function for Ctf4-dependent replisome tethering of Pol α was suggested by analogy with the rDNA-associated protein Tof2 and the Dna2 nuclease, which are recruited to the replisome by Ctf4 as part of a mechanism that preserves the integrity of the large array of rDNA repeats (Villa *et al*, 2016). As seen for Pol α, mutation of the CIP-boxes of Tof2 and Dna2 disrupted their interaction with Ctf4 but did not perturb DNA replication in yeast cells. Instead, the *tof2-4A* and *dna2-4A* alleles both led to a reduction in the size of chromosome 12 (Villa *et al*, 2016). However, this phenotype was not observed with *pol1-4A* cells (CE and KL, unpublished data), indicating that Ctf4 tethers Pol α to the replisome in order to support some other function.

A second possibility was indicated by the observation that Pol α co-purifies from yeast cell extracts with histone complexes that have been released from chromatin (Foltman *et al*, 2013). This suggested that Pol α might have an associated histone-binding activity that contributes to chromatin replication when Pol α is tethered to the replisome by Ctf4, analogous to the role of the histone-binding motif of the Mcm2 subunit of the CMG helicase (Ishimi *et al*, 1998; Foltman *et al*, 2013; Huang *et al*, 2015; Richet *et al*, 2015), which is required to preserve gene silencing near budding yeast telomeres (Foltman *et al*, 2013). Similar to the phenotype of *pol1-4A* cells, mutation of the conserved histone-binding motif of Mcm2 is not synthetic lethal with *mec1Δ* (Foltman *et al*, 2013) and thus does not perturb DNA replication in yeast cells.

If the *ADE2* marker gene is placed near a telomere in a wild-type yeast strain, it is expressed in some cells and repressed in others for many generations (Fig 2A), leading to colonies with white sectors (expressing *ADE2*) and red sectors (repression of *ADE2*). However, when a haploid strain with telomeric *ADE2* was crossed to the *mcm2-3A* histone-binding mutant and the resulting diploid strain was sporulated, the *mcm2-3A* progeny that inherited the marker gene produced pure white colonies (Foltman *et al*, 2013), in which all cells express sub-telomeric *ADE2* (Fig 2B). Strikingly, we found in similar experiments that *pol1-4A* is also defective in the maintenance of telomeric silencing (Fig 2C), and the same is true for *ctf4Δ* cells (Suter *et al*, 2004). Moreover, this phenotype was not produced by mutation of the CIP-box in other Ctf4 partners such as Tof2 or Dna2 (Fig EV1). These findings indicated that Ctf4-dependent tethering of Pol α to the replisome is required to preserve gene silencing near a telomere, but is not required for Pol α to fulfil its role in DNA synthesis. Consistent with this view, telomere length is normal in *pol1-4A* cells (Appendix Fig S2; compare *pol1-4A* and control), whereas telomere length increases as a consequence of defects in Pol α catalytic function (Adams & Holm, 1996), or following the displacement of primase from Pol α (Appendix Fig S2, *pol1-F1463A*).

Gene silencing in budding yeast cells also occurs at the silent mating-type genes on chromosome 3 (Haber, 2012). The *ADE2* marker gene is very strongly repressed when inserted at the *HMR* mating-type locus in control cells and therefore produces red

**Figure 1. Ctf4-dependent tethering of Pol alpha to the replisome is dispensable for efficacious DNA synthesis in budding yeast cells.**

A Control (YCE542) and *pol1-4A* (YCE544) cells were synchronized in G1-phase at 24°C then released into S-phase for the indicated times. DNA content was measured by flow cytometry throughout the experiment.

B In a similar experiment to that described above, samples from the G1-phase and 30′ S-phase timepoints were used to prepare cell extracts, from which the CMG helicase component Sld5 was isolated by immunoprecipitation. The associated replisome components were monitored by immunoblotting. The asterisk indicates a non-specific band in the anti-Pob3 immunoblot (corresponding to TAP-Sld5).

C Diploid cells of the indicated genotype were sporulated, and then, asci were dissected on rich medium. The image of the resultant tetrads was taken after 2 days growth at 30°C, and the genotype of each colony was determined by replica plating to selective media, indicating that the growth of *pol1-4A* is not affected by combination with *mec1Δ*.

D Equivalent analysis showing that *pol1-F1463A* is synthetic lethal with *mec1Δ*.

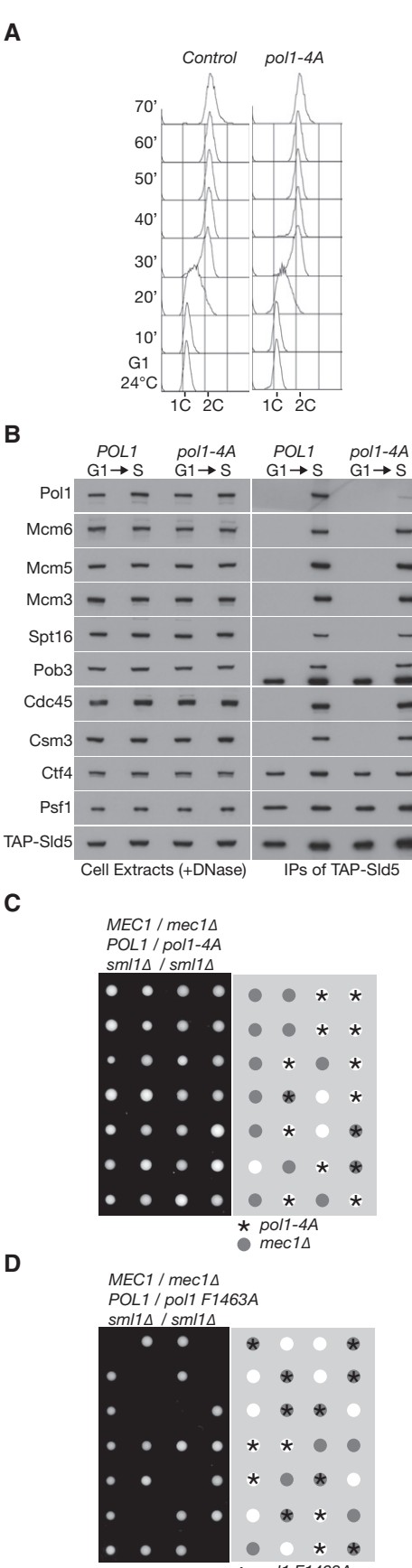

colonies that are only slightly paler than colonies of control cells (Fig 2D; *ADE2* replaces the silenced *MAT***a***2* gene and has the same orientation, designated *HMR::2EDA*). However, we found that *HMR-2EDA* is largely de-repressed in *mcm2-3A* and *pol1-4A* cells, which thus grew as very pale pink colonies (Fig 2E and F; note that the control *HMR-2EDA* cells sometimes produced dark pink rather than red colonies in this experiment, probably due to partial de-repression of *HMR-2EDA* in the parental diploid cells that were heterozygous for *mcm2-3A* or *pol1-4A*). These data indicate that the histone-binding activities of the replisome are important to preserve gene silencing at the mating-type loci, in addition to being important for telomeric silencing (Fig EV2B shows that the same is true in *HMR::ADE2* cells in which *ADE2* is inserted in the opposite orientation to *HMR::2EDA*, leading to weaker gene silencing).

Finally, we tested whether the *mcm2-3A* and *pol1-4A* mutations affected silencing of a *URA3* marker gene that had been inserted into one copy of the rDNA repeats on chromosome 12. Interestingly, gene silencing in the rDNA is mechanistically distinct from silencing at telomeres and the silent mating-type loci (Srivastava *et al*, 2016). The *URA3* gene was silenced when inserted into either the *NTS1* or *NTS2* sequences of an rDNA repeat, and silencing at *NTS1* was relieved in cells that lack the type I topoisomerase Top1 (Fig EV3), as reported previously (Huang *et al*, 2006). However, gene silencing at *NTS1* and *NTS2* persisted in *mcm2-3A* and *pol1-4A* cells. Therefore, although Mcm2 and replisome-tethered Pol α are both required to preserve gene silencing at telomeric and mating-type loci in budding yeast cells, they are dispensable for the mechanistically distinct phenomenon of rDNA silencing.

**The amino terminus of the DNA polymerase subunit of Pol α contains a conserved histone-binding motif**

The ability of Pol α to interact in yeast cell extracts with chromatin-derived histone complexes might reflect a novel histone-binding activity in Pol α, but could also be due to the previously described association of Pol α with the histone chaperone FACT (Miles & Formosa, 1992). Consistent with the former possibility, we noticed that the extended amino-terminal tail of the Pol α catalytic subunit

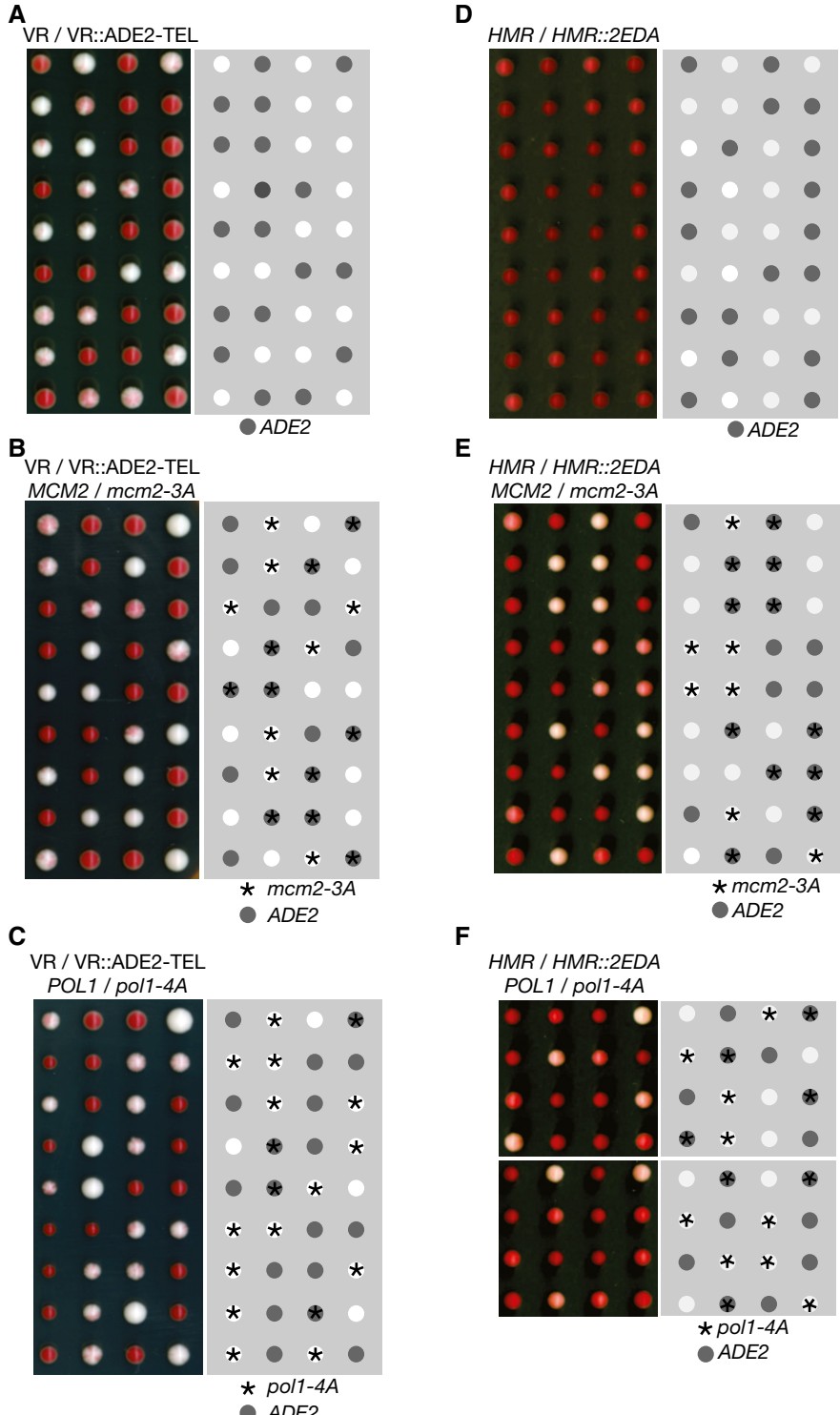

**Figure 2.  Displacement of Pol alpha from the replisome leads to a loss of gene silencing at sub-telomeric or mating-type loci.**

A–C    Tetrad analysis of meiotic progeny of the indicated diploids (heterozygous for sub-telomeric *ADE2*), processed as described in Materials and Methods. Sectored colonies represent epigenetic variation in the expression of the *ADE2* marker at the sub-telomeric location. White colonies indicate expression of *ADE2* in all cells, and red colonies lack the *ADE2* marker (as denoted in the right-hand panels). We examined a total of 19 *mcm2-3A VR::ADE2-TEL* colonies and 11 *pol1-4A VR::ADE2-TEL* colonies.

D–F    Equivalent analysis of the meiotic progeny of diploid cells that were heterozygous for an insertion of the *ADE2* marker at *HMR* locus on chromosome 3 (denoted *HMR::2EDA* since the orientation of the *ADE2* marker was such that the promoter was distal to the *HMR-E* silencer element). We examined a total of 19 *mcm2-3A HMR::2EDA* colonies and 12 *pol1-4A HMR::2EDA* colonies.

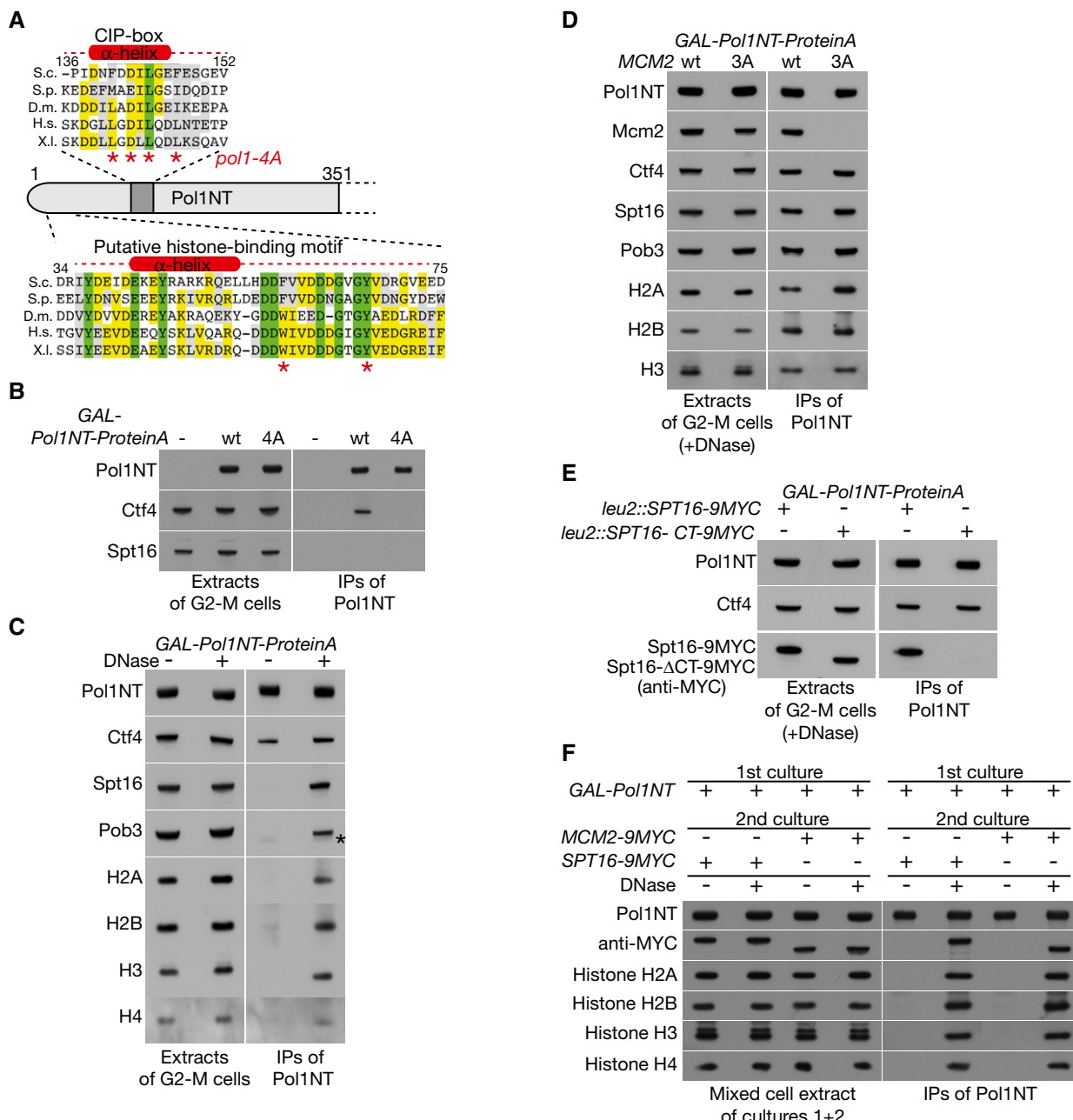

**Figure 3. The amino terminus of Pol1 binds chromatin-derived histone complexes, together with Mcm2 and FACT.**

A   Sequence alignment indicating the location of the CIP-box and a newly identified putative histone-binding motif in the amino-terminal tail of budding yeast Pol1. Asterisks under the CIP-box indicate residues mutated in the *pol1-4A* allele, whereas those under the novel histone-binding motif denote two conserved aromatic residues separated by eight amino acids, as in the histone-binding motif of Mcm2.

B   Control cells (YSS3), cells expressing ProteinA-tagged Pol1NT (YCE39) and cells expressing ProteinA-tagged Pol1NT-4A (YCE217) were synchronized in G2-M phase at 30°C, before the isolation of the ProteinA-tagged Pol1 tails on IgG beads. Pol1NT interacts with Ctf4, dependent upon the CIP-box, but does not interact with FACT (for which the Spt16 subunit is shown).

C   Cells expressing ProteinA-tagged Pol1NT (YCE39) were grown as above, but the resultant extracts were then incubated +/−DNase, before isolation of Pol1NT. The associated proteins were monitored by immunoblotting—the asterisk in the Pob3 blot indicates faint cross-reactivity with ProteinA-Pol1NT.

D   Analogous experiment comparing expression of GAL-Pol1NT in control cells (YCE39) or *mcm2-3A* cells (YCE238). After release of histones from chromatin by DNase treatment, the co-purification of Mcm2 with Pol1NT depends on the H3-H4 binding motif of Mcm2, which is mutated in *mcm2-3A*.

E   Similar experiment comparing expression of GAL-Pol1NT in cells with a second copy of *SPT16* at the *leu2* locus, expressing either wild-type Spt16 (YCE248) or Spt16 with a small deletion at the carboxyl terminus (YCE250) that abolishes interaction of FACT with H3-H4 tetramers.

F   Cultures of *GAL-Pol1NT* (YCE39), *MCM2-9MYC* (YMP154-1) and *SPT16-9MYC* (YMP177-1) were arrested in G2-M phase at 30°C, and then, equal volumes were mixed as indicated. Cell extracts were prepared, and Pol1NT was isolated on IgG-coated magnetic beads. Release of histones from chromatin by DNase treatment allowed Pol1NT from the first culture to form ternary complexes containing not only histones but also Mcm2-9MYC or Spt16-9MYC from the second cultures.

in eukaryotes contains a motif that resembles the histone-binding element in Mcm2 (Fig 3A), in that it contains two conserved aromatic residues that are separated by eight amino acids and are flanked by conserved acidic residues, adjacent to a predicted alpha helix (Foltman *et al*, 2013). However, the two motifs also have important differences, since the orientation of the alpha helix and the adjacent pair of aromatic residues is reversed between Mcm2 and Pol α, and the Pol1 tail has an additional pair of conserved tyrosines that overlap with the predicted alpha helix.

To test the ability of the amino-terminal tail of budding yeast Pol1 to associate with histone complexes and FACT, we expressed the first 351 amino acids of Pol1 in yeast cells as a fusion to Protein A, and then isolated the fusion protein from cell extracts, with or without prior DNase treatment to release histone complexes from chromatin. As shown in Fig 3B, the amino-terminal fragment of Pol1 (hereafter termed Pol1NT) associated specifically with Ctf4, dependent upon the integrity of the CIP-box (Fig 3B, compare Pol1NT-4A with wt Pol1NT), but did not associate with FACT, which must instead bind to some other region of Pol α. However, upon digestion of genomic DNA to release histone complexes from chromatin, Pol1NT co-purified not only with histones but also with FACT in yeast cell extracts (Fig 3C, +DNase; Appendix Fig S3 shows the salt sensitivity of the observed complexes). These findings suggested that Pol1NT shares the ability of Mcm2 to associate with histone complexes that have been released from chromatin and also indicated that the same histone-containing complexes can be bound by FACT.

To explore this further, we expressed Pol1NT (Pol1 1–351) and Mcm2NT (Mcm2 1–200) in yeast cells as fusions to Protein A and also included Protein A-fused Mcm4NT (Mcm4 1–186) as a negative control. We isolated the fusion proteins by immunoprecipitation

from yeast extracts with or without DNase treatment, before analysis of the associated factors by mass spectrometry. As summarized in Table 1, the treatment of yeast extracts with DNase greatly stimulated the detectable association of both Pol1NT and Mcm2NT, but not Mcm4NT, with histones and FACT. In addition, Mcm2 (but not Mcm4) was greatly enriched in the Pol1NT immunoprecipitates after DNase treatment of extracts. Incidentally, we found that the Tra1 subunit of the SAGA and NuA4 histone acetyltransferases was also enriched in the immunoprecipitates of Pol1NT and Mcm2NT after DNase treatment, suggesting that Tra1 might also bind to histone complexes released from chromatin, presumably on distinct sites to Mcm2NT and Pol1NT.

As expected, we found that Pol1NT interacted specifically with Ctf4 regardless of DNase treatment. Moreover, Pol1NT interacted with the Tel1 checkpoint kinase in a similar manner to Ctf4 (Table 1), indicating that Tel1 is a novel partner of Pol1NT. However, subsequent experiments indicated that neither Ctf4 (Fig EV4A) nor Tel1 (Fig EV4B) was required for Pol1NT to associate with histone complexes released from chromatin. Consistent with this, *tel1Δ* was not associated with a loss of telomeric silencing (Fig EV1), though it did reduce telomere length (Appendix Fig S2), as previously reported (Lustig & Petes, 1986; Greenwell *et al*, 1995).

The presence of Mcm2 in the Pol1NT immunoprecipitates was dependent not only upon the release of histones from chromatin by DNase treatment, but also required the integrity of the conserved histone-binding motif in the Mcm2 tail (Fig 3D, *mcm2-3A*). Similarly, the co-purification of FACT with Pol1NT, upon release of histones from chromatin, was abolished by mutation of the carboxyl terminus of the Spt16 subunit (Fig 3E and Appendix Fig S4), corresponding to a small truncation that was previously shown

**Table 1. Mass spectrometry analysis of proteins co-purifying with Pol1NT, Mcm2NT or Mcm4NT, +/−DNase treatment to release histone complexes from chromatin.**

| Identified protein | MS analysis of IPs from G2-M cell extracts (spectral counts) | | | | |
|---|---|---|---|---|---|
| | Pol1 (1–351) +DNase | Pol1 (1–351) −DNase | Mcm2 (1–200) +DNase | Mcm2 (1–200) −DNase | Mcm4 (1–186) +DNase |
| Pol1NT | 1,570 | 854 | 7 | 0 | 0 |
| Mcm2 | 436 | 0 | 258 | 218 | 6 |
| Mcm4 | 38 | 11 | 14 | 14 | 681 |
| Ctf4 | 938 | 564 | 0 | 0 | 0 |
| Tel1 | 407 | 229 | 18 | 0 | 0 |
| Histone H2A | 95 | 0 | 46 | 0 | 16 |
| Histone H2B | 441 | 3 | 140 | 3 | 33 |
| Histone H3 | 275 | 2 | 178 | 5 | 13 |
| Histone H4 | 287 | 4 | 275 | 4 | 28 |
| Spt16 | 2,255 | 53 | 1,558 | 5 | 201 |
| Pob3 | 812 | 8 | 486 | 0 | 46 |
| Tra1 | 600 | 32 | 1,105 | 44 | 86 |
| Spt5 | 396 | 22 | 38 | 3 | 26 |

The indicated protein fragments were expressed in G2-M-arrested cells and then isolated by immunoprecipitation on IgG-coated magnetic beads. The resultant material was resolved in a 4–12% gradient gel, which was then stained with colloidal Coomassie blue, before each lane was cut into 40 bands for mass spectrometry analysis. The table summarizes the spectral counts that were detected for each factor that specifically co-purified with the indicated fragment.

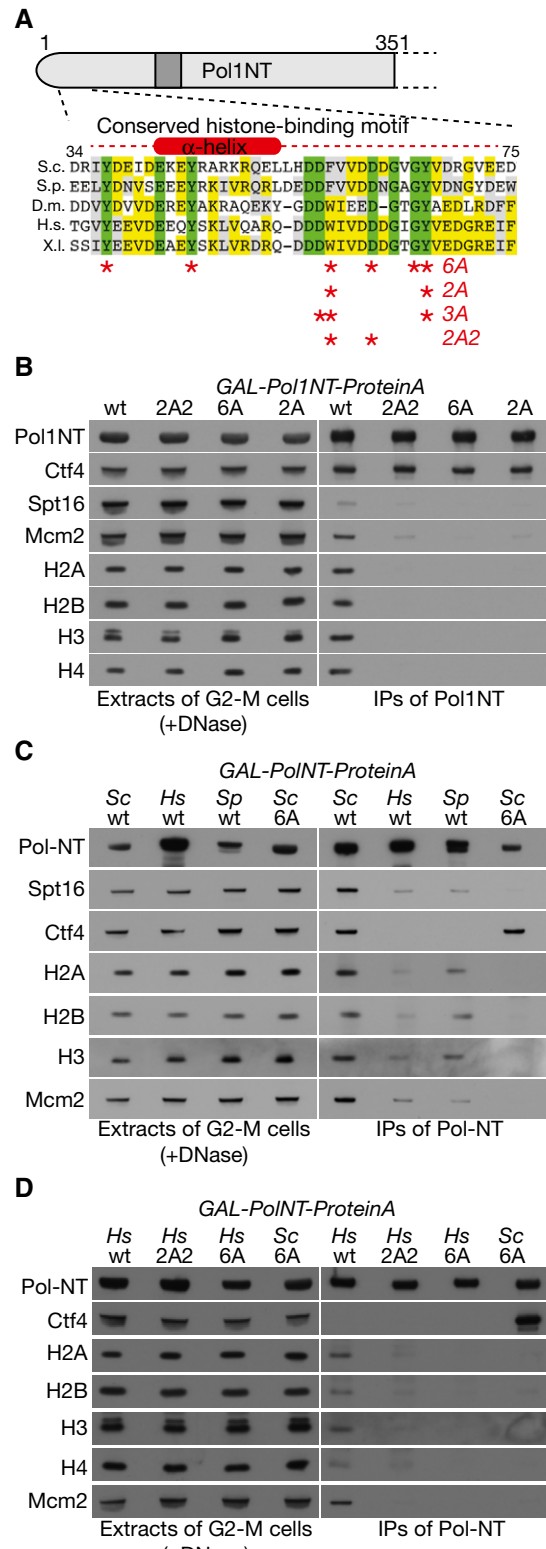

**Figure 4.  A conserved motif at the amino terminus of Pol1 is required to bind chromatin-derived histone complexes, together with Mcm2 and FACT.**

A  Asterisks correspond to the residues in the newly identified histone-binding motif that were mutated to create the indicated alleles.

B  The amino-terminal tails of Pol1-2A (YCE1084), Pol1-2A2 (YCE232) and Pol1-6A (YCE226) still interact with Ctf4, but are unable to associate with histones released from chromatin, and do not co-purify with FACT.

C  The amino-terminal tail of the DNA polymerase alpha catalytic subunit from fission yeast (YCE378; Sp = *Schizosaccharomyces pombe*) and human (YCE407; Hs = *Homo sapiens*) bind specifically to histone complexes derived from budding yeast chromatin, by comparison with the Pol1-6A budding yeast tail with a mutated histone-binding site.

D  Mutation of conserved residues in the histone-binding motif (YCE425 expresses the 2A2 allele and YCE435 corresponds to the 6A allele) kills the ability of Hs-Pol1NT to associate with histone complexes derived from budding yeast chromatin.

complexes from DNA, we mixed a yeast culture expressing Pol1NT with cells expressing either Mcm2-9MYC or Spt16-9MYC, before making mixed cell extracts. In the absence of DNase treatment, Pol1NT from the first cell culture did not interact with either Mcm2-9MYC or SPT16-9MYC from the second cultures (Fig 3F, −DNase). In contrast, DNase treatment of the cell extracts allowed Pol1NT from the first culture not only to associate *in vitro* with histones, but also to co-purify with tagged Mcm2 or Spt16 from the second culture (Fig 3F, +DNase). Overall, these findings indicated that Pol1NT is able to interact *in vitro* with chromatin-derived histone complexes, which can also be co-chaperoned by Mcm2 and FACT.

Mutation to alanine of six conserved residues in Pol1NT (Y37, Y45, F58, D62, G66 and Y67), within the region of similarity with the histone-binding motif of Mcm2, greatly reduced the ability of Pol1NT to co-purify with histones, FACT and Mcm2, without affecting the interaction of Pol1NT with Ctf4 (Fig 4A and B, and Appendix Fig S5A and B). Further analysis of subsets of these mutations (Appendix Fig S5C) highlighted the importance of the two conserved aromatic residues F58 and Y67 that are located downstream of the predicted alpha helix (Fig 4A and B, Pol1-2A), along with neighbouring acidic amino acids (Fig 4A and B, Pol1-2A2 containing the F58A and D62A mutations). In addition, the pair of conserved tyrosines that overlap with the alpha helix also contributed to the observed histone-binding activity (Appendix Fig S5, Y37A Y45A).

To test whether the corresponding region of Pol α from other species is able to bind to chromatin-derived histone complexes, we expressed the fission yeast and human equivalents of Pol1NT in budding yeast cells, as fusions to Protein A. Neither of the fusion proteins bound to budding yeast Ctf4, probably due to the evolutionary divergence of the human and fission yeast CIP-box sequences. However, both proteins co-purified with yeast histones, FACT and Mcm2, reflecting the high conservation of histone sequences amongst diverse eukaryotic species (Fig 4C; note that the binding of fission yeast and human Pol α tails to budding yeast histone complexes was weaker than seen with wild-type ScPol1NT, but was still clearly specific in comparison with ScPol1NT-6A). Moreover, the ability of HsPolα-NT to bind to yeast-chromatin-derived histone complexes, together with Mcm2 and FACT, was abrogated by mutations in conserved residues within the novel

to abolish the association of FACT with H3–H4 tetramers (Tsunaka *et al*, 2016). These findings indicated that Pol1NT, Mcm2NT and FACT are able to bind to different regions of the same histone complexes that are released from chromatin. To confirm that these interactions can indeed occur *in vitro* upon release of histone

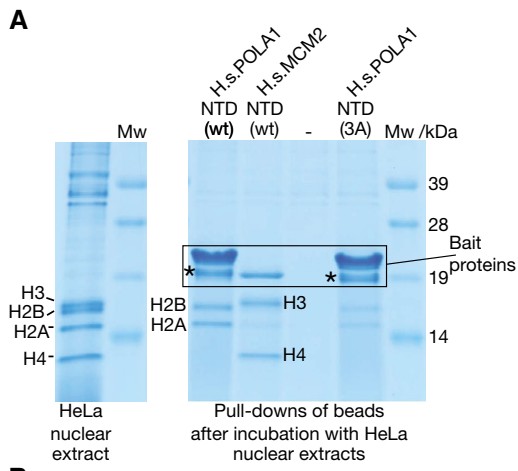

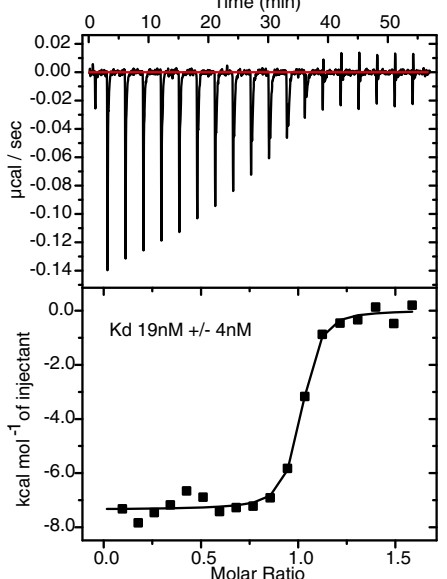

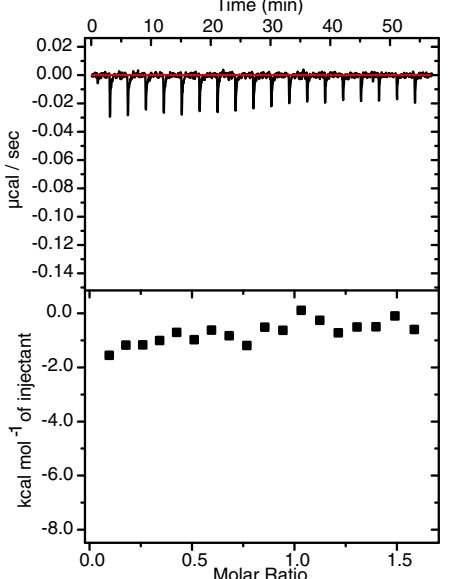

**Figure 5. The histone-binding motif in the amino-terminal tail of the DNA polymerase subunit of Pol α is specific for histones H2A-H2B.**

A   Recombinant versions of the amino-terminal tail of human POLA1 (amino acids 1–110 of the polymerase subunit of Pol α) or human MCM2 (residues 43–160) were purified from *Escherichia coli* as fusions to a Twin-Strep-tag. The Strep-Tactin beads carrying the purified tails were then incubated with a high-salt extract of HeLa nuclei, before the beads were washed and the bound proteins eluted in *d*-Desthiobiotin buffer. The eluted proteins were revealed by SDS–PAGE and Coomassie staining, and relevant bands were identified by mass spectrometry and confirmed by immunoblotting. The asterisk denotes a degradation product of the POLA1-NTD.

B   The interaction of recombinant H.s. POLA1-NTD with a complex of recombinant histones H2A-H2B was monitored by isothermal titration calorimetry. The experiment was performed three times, and the figure shows a representative example.

C   Similar analysis of the interaction of H.s. POLA1-NTD (6A mutant) with a complex of recombinant human histones H2A-H2B.

histone-binding motif (Fig 4D). Overall, these findings indicated that Pol1NT contains an evolutionarily conserved histone-binding motif that is capable of binding to chromatin-derived histone complexes, on distinct sites to those bound by Mcm2 and FACT.

To explore which histones are bound by the novel motif in Pol α, we took advantage of the fact that HeLa nuclei contain high concentrations of free histones that are available to bind to histone chaperones, whereas in budding yeast cells, the non-nucleosomal pool of free histones is extremely low. We incubated extracts of HeLa nuclei with bead-bound versions of recombinant HsMCM2-NT (amino acids 43–160 of human MCM2), wild-type HsPOLA1-NT (amino acids 1–110 of human POLA1), or HsPOLA1-3A-NT (Fig 4A shows the location of the 3A mutations within the conserved histone-binding motif of HsPOLA1-NT). After recovery of the bead-bound versions of the HsMCM2-NT and HsPOLA1-NT tails, the major associated proteins were visualized by Coomassie staining (Fig 5A) and then identified by mass spectrometry and immunoblotting (AD, JM and LP, unpublished data). HsMCM2-NT bound specifically to histones H3-H4, as observed previously (Huang *et al*, 2015; Richet *et al*, 2015). In contrast, HsPOLA1-NT bound preferentially to histones H2A-H2B (Fig 5A), and the observed association was diminished by the 3A mutations of conserved residues, including those that we showed to be important for the association of yeast Pol1NT with chromatin-derived histone complexes. To confirm and quantify the interaction of HsPOLA1-NT with histones H2A-H2B, we monitored the interaction of recombinant HsPOLA1-NT and H2A-H2B by isothermal titration calorimetry (ITC), as described in Materials and Methods. These experiments indicated that HsPOLA1-NT bound with 1:1 stoichiometry to an H2A-H2B heterodimer, with a dissociation constant of 19 nM (Fig 5B). Moreover, binding was abolished by mutation of six conserved residues in the histone-binding motif (equivalent to the budding yeast Pol1-6A mutant), confirming the specificity of the observed interaction (Fig 5C). Overall, therefore, these data demonstrate that the amino-terminal tail of the DNA polymerase subunit of human Pol α contains a conserved motif that binds specifically to histones H2A-H2B. Moreover, our data indicate that Pol α, Mcm2 and FACT are able to bind simultaneously to distinct sites on the surface of chromatin-derived histone complexes that contain all four yeast histones, after release of the latter from DNA.

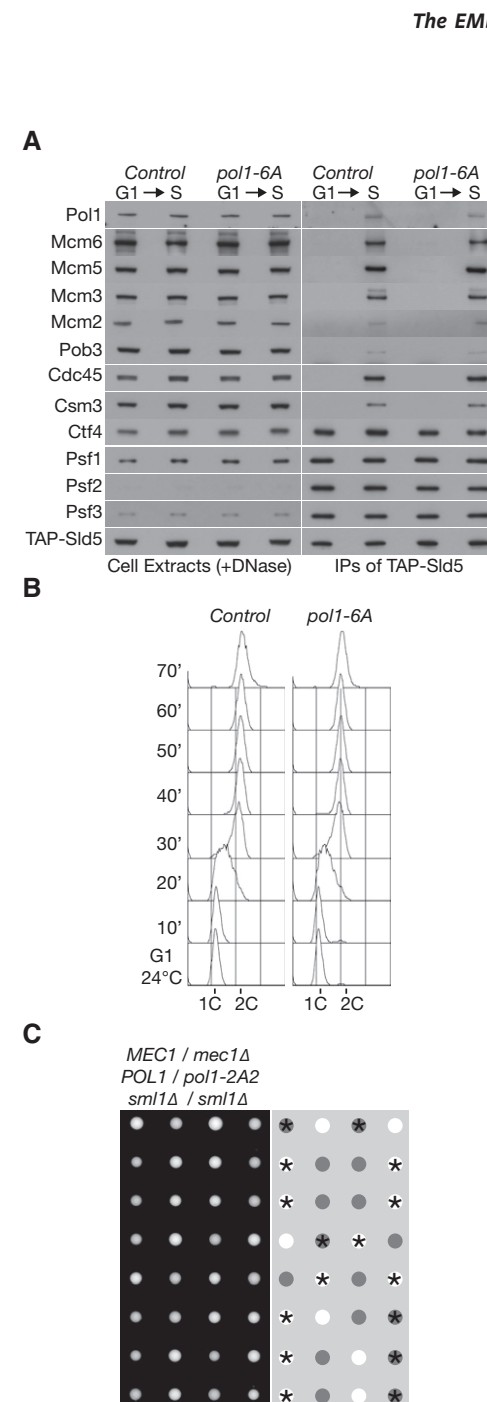

**Figure 6.  The histone-binding motif in the amino-terminal tail of budding yeast Pol1 is dispensable for replisome tethering of Pol alpha and for efficacious DNA synthesis.**

A    Control (YCE542) and *pol1-6A* (YCE546) cells were synchronized in G1-phase at 24°C and then released into S-phase for 30′, before replisome assembly was monitored as above, via immunoprecipitation of the Sld5 subunit of the CMG helicase.

B    The kinetics of DNA synthesis for the *pol1-6A* mutant were monitored by flow cytometry, in the same experiment shown above in Fig 1A (alongside control and *pol1-4A*). Therefore, the control sample in this panel is identical to that shown in Fig 1A.

C, D    The growth of *pol1-2A2* and *pol1-6A* was not affected by combination with *mec1Δ*.

## Mutation of the histone-binding motif in budding yeast Pol1 disrupts gene silencing without affecting DNA synthesis or the tethering of Pol α to the replisome

To investigate the *in vivo* role of the conserved histone-binding motif in budding yeast Pol1NT, we introduced the *pol1-6A* or *pol1-2A2* mutations into the endogenous *POL1* locus in yeast cells, in order to compare the phenotypes of mutating the histone-binding motif of Pol1 with the *pol1-4A* allele that displaces Pol α from Ctf4 in the replisome. Replisome assembly was normal in *pol1-6A* (Fig 6A and Appendix Fig S6A) and *pol1-2A2* cells (Appendix Fig S6C), including the tethering of Pol α to the CMG helicase, and the kinetics of DNA replication were indistinguishable from control cells (Fig 6B and Appendix Fig S6D). Moreover, neither *pol1-2A2* nor *pol1-6A* caused sensitivity to the replication inhibitor hydroxyurea (Fig EV5), and the viability of *pol1-2A2* and *pol1-6A* was independent of the Mec1 checkpoint kinase (Fig 6C and D). These data indicate that the ability of Pol1NT to bind to chromatin-derived histone complexes is dispensable for efficient DNA synthesis during chromosome replication in budding yeast cells. Importantly, however, gene silencing near a telomere and at the HMR mating-type locus was defective in *pol1-2A2* and *pol1-6A* cells (Figs 7A–D and EV2D and E), whereas rDNA silencing was not abolished (Fig EV3). Therefore, the phenotype of a replisome in which Pol α has a defective histone-binding motif resembles the phenotype of displacing Pol α from the replisome (*pol1-4A*, described above). These findings indicate that Ctf4-dependent replisome tethering of Pol α in budding yeast is dispensable for DNA synthesis at replication forks and instead contributes to histone processing and gene silencing.

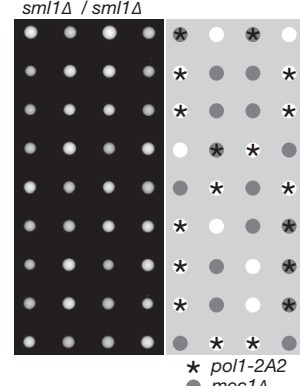

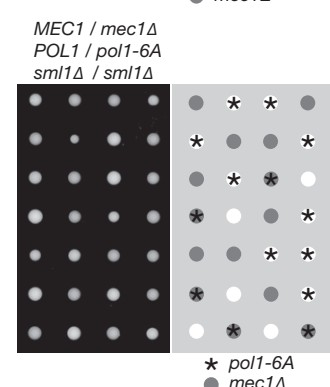

## Discussion

Long before the identification of the CMG helicase first made possible the characterization of the eukaryotic replisome, work with eukaryotic DNA viruses suggested that the tethering of Pol α to the replicative helicase might be an important feature of replisome assembly in eukaryotic cells. For example, Pol α was shown to interact directly with the replicative helicases of SV40 (Dornreiter *et al*, 1990) and bovine papillomaviruses (Park *et al*, 1994; Bonne-Andrea *et al*, 1995), as part of a minimal replisome in which the viral helicase associates with eukaryotic Pol α and topoisomerase 1, but not with the leading strand polymerase or with other eukaryotic replisome components that interact with the CMG helicase. However,

**A**

VR / VR::ADE2-TEL
*POL1 / pol1-2A2*

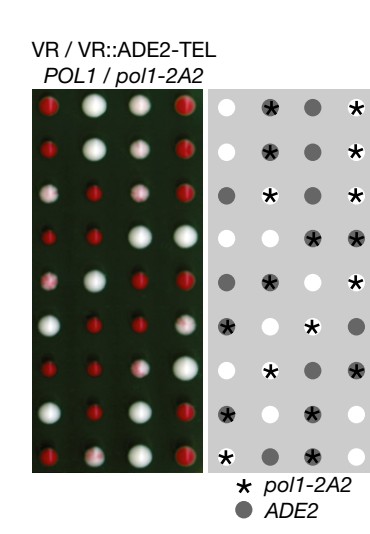

✴ *pol1-2A2*
● *ADE2*

**B**

VR / VR::ADE2-TEL
*POL1 / pol1-6A*

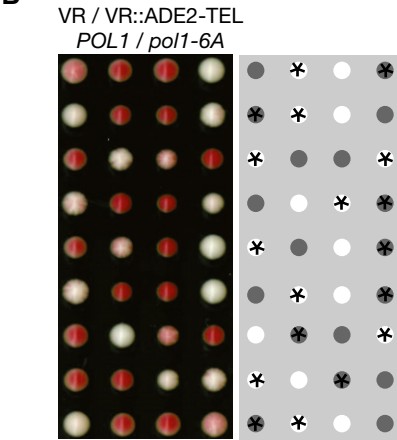

✴ *pol1-6A*

**C**

*HMR / HMR::2EDA*
*POL1 / pol1-2A2*

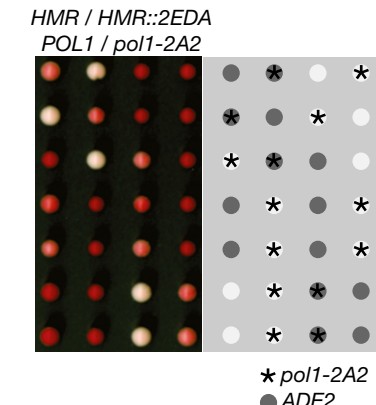

★ *pol1-2A2*
● *ADE2*

**D**

*HMR / HMR::2EDA*
*POL1 / pol1-6A*

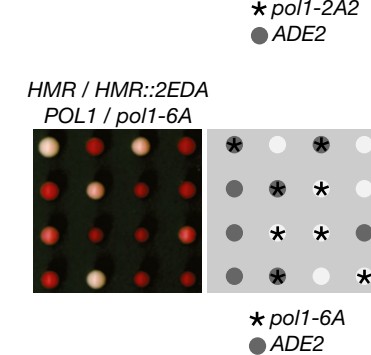

★ *pol1-6A*
● *ADE2*

Figure 7.  The histone-binding activity of Pol1 is required for gene
silencing at sub-telomeric and mating-type loci.

A, B   The indicated strains were analysed as in Fig 2A and B. The *pol1-2A2* and
        *pol1-6A* alleles are defective in sub-telomeric gene silencing. We
        examined a total of 15 *pol1-2A2 VR::ADE2-TEL* colonies and 19 *pol1-6A VR::
        ADE2-TEL* colonies.
C, D   Mating-type silencing was analysed as in Fig 2D–F. Silencing was
        found to be defective in the *pol1-2A2* and *pol1-6A* strains. We
        examined a total of 15 *pol1-2A2 HMR::2EDA* colonies and 13 *pol1-6A
        HMR::2EDA* colonies.

tethering of Pol α to the eukaryotic replisome is more complex than for eukaryotic viruses, since it is largely dependent upon the Ctf4 protein (Zhu *et al*, 2007; Gambus *et al*, 2009; Tanaka *et al*, 2009a), in addition to a residual interaction of Pol α with the CMG helicase (Appendix Fig S1, Sengupta *et al*, 2013). This role for Ctf4 was first described for budding yeast but is likely to be conserved broadly in other species, since Ctf4 from fission yeast, human and frog has also been reported to interact with Pol α (Zhu *et al*, 2007; Tanaka *et al*, 2009b; Kilkenny *et al*, 2017).

Recent *in vitro* studies of a reconstituted budding yeast replisome system suggested that Pol α synthesizes RNA-DNA hybrids in a distributive fashion at eukaryotic DNA replication forks, at least on a naked DNA template, implying that replisome tethering of Pol α by Ctf4 does not on its own ensure efficient priming (Yeeles *et al*, 2017). Consistent with this idea, Ctf4 is dispensable for synthesis of the lagging strand by the reconstituted yeast replisome (Yeeles *et al*, 2015). Efficient priming by Pol α might instead depend on functional interactions with the CMG helicase (Georgescu *et al*, 2015; Taylor & Yeeles, 2018), the single-strand DNA-binding protein RPA (Dornreiter *et al*, 1992; Nasheuer *et al*, 1992; Braun *et al*, 1997), the DNA template or even chromatin (Kurat *et al*, 2017).

Our data indicate that Ctf4-dependent tethering of Pol α and other factors to the eukaryotic replisome has evolved to serve functions that go beyond DNA synthesis. Strikingly, CIP-box mutations in the known partners of budding yeast Ctf4 have no impact on DNA synthesis, but instead lead to phenotypes that reflect the roles of Ctf4 in other aspects of chromosome replication, such as sister chromatid cohesion (Samora *et al*, 2016), preserving the length of chromosome 12 that contains the rDNA (Villa *et al*, 2016), or the maintenance of gene silencing (this study). These findings highlight the role of Ctf4 as a hub in the eukaryotic replisome that coordinates diverse processes with DNA synthesis. Moreover, these data emphasize the notion that replisome assembly in eukaryotes evolved not simply to facilitate timely and efficient DNA synthesis, but also to promote chromosome replication in all its complexity, by coupling the CMG helicase to other processes that can only be realized as part of a dynamic multi-protein assembly.

Our data indicate that the amino-terminal tail of the Pol α DNA polymerase subunit contains a novel histone-binding activity. Unexpectedly, given that the identification of the motif was based on its partial resemblance to the H3-H4 binding motif of human Mcm2, we show that the histone-binding motif of human Pol α binds to histones H2A-H2B. The Pol α histone-binding motif appears to be more complex than the equivalent element in Mcm2-NT, and it will be fascinating in future studies to investigate the structural basis for histone binding by Pol α-NT, for both the human and yeast proteins.

Despite the differences in histone binding by Mcm2 and Pol α, mutation of the histone-binding motif in either protein produces very similar phenotypes in budding yeast cells. Firstly, DNA synthesis is not impaired (Fig 6 and Appendix Fig S6D), even when both motifs are mutated simultaneously (Fig EV5). Instead, the histone-binding activities of both Mcm2 and Pol1 are required to preserve gene silencing in telomeric and mating-type loci (Figs 7A–D and EV2), and yet are largely dispensable for gene silencing in the rDNA repeats (Fig EV3).

Both at telomeres and at the silent mating-type genes, a complex of the silent information repressor (SIR) proteins Sir2-Sir3-Sir4 is first recruited to specific sites and then spreads to mediate gene silencing over a distance of several kilobases (Haber, 2012; Wellinger & Zakian, 2012). The initial recruitment of Sir2-Sir3-Sir4 at telomeres is mediated by the Rap1 protein bound to the telomeric DNA repeats (Wellinger & Zakian, 2012), whereas at the silent mating-type loci, the origin recognition complex promotes the recruitment of the Sir1 protein (Haber, 2012), which in turn loads the first Sir2-Sir3-Sir4 complex. Once recruited to telomeres and to mating-type silencers, Sir2 deacetylates lysine residues in the amino-terminal tails of histones H3 and H4, within the adjacent nucleosomes. Deacetylation of H4K16 is particularly important and promotes the stable binding of Sir2-Sir3-Sir4, initiating an iterative process by which Sir2-Sir3-Sir4 are able to spread away from the initial site of recruitment, driven by successive rounds of histone deacetylation by Sir2 and the consequent and stable binding of Sir2-3-4. Across the region to which the Sir proteins are bound, they mediate gene silencing by inhibiting the recruitment of transcription factors and tethering the loci to the nuclear periphery.

Our data indicate that Mcm2 and Ctf4-tethered Pol α are required to preserve the functional integrity of chromatin regions that are silenced by Sir2-3-4. We suggest that the replisome's histone-binding activities help to retain parental histone complexes after their initial displacement from DNA at replication forks, so that the histones can be re-deposited close to their original location. If the histone-binding activity of the replisome is abrogated by mutation, a higher proportion of parental histones from other loci, with acetylated H4 and H3 tails, might then be deposited onto nascent DNA across the previously silenced regions, thus leading to displacement of Sir2-3-4 and the disruption of gene silencing. In such circumstances, it is likely that there is insufficient time before the next S-phase for Sir2-Sir3-Sir4 to spread once again across the formerly silenced region, thus explaining the gene silencing defect in the histone-binding mutants.

Gene silencing in the rDNA is considerably more complex than at the telomeric and silent mating-type loci (Huang & Moazed, 2003; Huang *et al*, 2006; Srivastava *et al*, 2016). Although the Sir2 histone deacetylase still plays an important role, both Sir3 and Sir4 are dispensable for rDNA silencing. This is because Sir2 is recruited to the rDNA by the RENT complex, which in turn associates with Fob1 that is bound to the replication fork barrier (Srivastava *et al*, 2016). Moreover, the Tof2 protein is also recruited by Fob1 and supports a Sir2-independent mechanism of silencing. The latter pathway is understood poorly, but our data suggest that rDNA silencing does not require the faithful retention of parental histone complexes close to their original location during the passage of a DNA replication fork, implying that rDNA silencing can be re-established *de novo* after the passage of a replication fork, before the S-phase of the subsequent cell cycle.

When the parental DNA duplex is unwound at eukaryotic DNA replication forks, parental histones are released transiently into the nucleoplasm before re-associating with nascent DNA (Radman-Livaja *et al*, 2011; Kurat *et al*, 2017). Very little is known about the mechanisms of parental histone transfer during chromosome replication, but past work showed that an intact tetramer of parental histones H3-H4 is preserved on nascent DNA after replication (Prior *et al*, 1980; Yamasu & Senshu, 1990; Vestner *et al*, 2000; Katan-Khaykovich & Struhl, 2011), and a recent study highlighted a role for the eukaryotic replisome in ensuring that histones H3-H4 are retained close to their original genomic location after the passage of a DNA replication fork (Madamba *et al*, 2017).

We favour a model in which the unwinding of the initial segment of nucleosomal DNA by the CMG helicase allows FACT to use its multiple histone-binding activities to destabilize histone-DNA interactions within the remainder of the nucleosome. Firstly, FACT detaches a dimer of H2A-H2B from the octamer (Belotserkovskaya *et al*, 2003; Tsunaka *et al*, 2016), and binds to the H3-H4 tetramer of the resulting histone hexamer (Tsunaka *et al*, 2016), destabilizing histone-DNA interactions and promoting the continued unwinding of nucleosomal DNA by CMG (Kurat *et al*, 2017). After the DNA has been stripped away, the histone hexamer is then chaperoned and retained in the vicinity of the replication fork (potentially together with the detached H2A-H2B dimer), by the binding of multiple replisome components including FACT, Mcm2 and Pol α, before the local re-deposition of the parental histones onto nascent DNA behind the replisome (Kurat *et al*, 2017). The latter process is coupled to the assembly of additional nucleosomes from newly synthesized histones, and the rapid exchange of H2A-H2B in nascent chromatin then leads to the observed situation, whereby parental H3-H4 tetramers are inherited stably during chromosome replication, whereas H2A-H2B are considerably more dynamic.

Whereas human FACT and Mcm2 are known to bind to adjacent non-overlapping sites on the H3-H4 tetramer (Huang *et al*, 2015; Richet *et al*, 2015; Tsunaka *et al*, 2016), our data indicate that human Pol α binds to H2A-H2B, and all three replisome components from budding yeast are able to bind simultaneously to the same chromatin-derived histone complexes (perhaps to a histone hexamer, after FACT detaches an H2A-H2B dimer from the octamer). The ability of multiple replisome components to bind to different surfaces on the same nucleosome-derived histone complexes creates the possibility of a relay mechanism, whereby parental histone complexes are passed from one replisome component to another before deposition on nascent DNA immediately behind the replisome. Alternatively, simultaneous binding by multiple replisome components might be required to chaperone the parental histone complexes that have been released from DNA.

Consistent with our proposed role for multiple replisome components in the processing of parental histones, recent structural work indicates that both Ctf4-Pol α and the amino-terminal tail of Mcm2 are located at the front face of the yeast replisome (Georgescu *et al*, 2017; Douglas *et al*, 2018) and thus are orientated towards the site from which parental histones are released during DNA unwinding at replication forks. The precise location of FACT in the replisome has yet to be determined, but FACT is part of the replisome progression complex that assembles around the CMG helicase (Gambus *et al*, 2006). It is likely that the replisome can accommodate at least one additional copy of FACT, tethered to Pol α (Miles & Formosa, 1992),

plus a further chaperone for histone complexes released from chromatin, namely the Dpb3-Dpb4 subunits of Pol ε (Tackett *et al*, 2005). As seen upon mutation of the histone-binding activity of Mcm2 or Pol α, cells lacking Dpb3–Dpb4 are viable in the absence of the Mec1 checkpoint kinase (Appendix Fig S7) and are defective in sub-telomeric silencing (Iida & Araki, 2004) but not rDNA silencing (Fig EV3).

In the future, it will be of particular interest to study organisms with more complex epigenetic regulation of gene expression than *Saccharomyces cerevisiae*, since such studies might reveal more complex phenotypes upon mutation of the replisome's histone-binding activities. For example, we note that a recent study of the plant orthologue of Ctf4 (Zhou *et al*, 2017) found that it was required to preserve gene silencing that is dependent upon trimethylation of histone H3K27—an epigenetic mark not found in budding yeast. Moreover, previous work showed that mutations in fission yeast Pol alpha also lead to defects in gene silencing, reflecting displacement from chromatin of the Swi6 chromodomain protein that binds to H3K9Me—another mark that is absent in *S. cerevisiae* (Nakayama *et al*, 2001). Nevertheless, the budding yeast replisome provides an important model system with which to identify histone-binding activities that are highly conserved in other species, thus defining motifs and residues that can be studied subsequently in metazoa or plants.

## Materials and Methods

### Yeast strains and growth

All *S. cerevisiae* strains used in this study are listed in Appendix Table S1. Cells were grown in rich medium (1% yeast extract, 2% peptone) supplemented with 2% glucose (YPD), 2% raffinose (YPRaff) or 2% galactose (YPGal). G1 phase samples were obtained by arresting exponentially growing cells ($7 \times 10^6$/ml) with 7.5 μg/ml α-factor (Pepceuticals) until 90% were unbudded and S samples by releasing G1-arrested cells for 30′ (at 24°C) into fresh medium lacking mating pheromone. To arrest cells in G2-M phase, a mid-exponential culture was treated with 5 μg/ml nocodazole (Sigma-Aldrich, M1404) for 2 h.

### Dilution spotting

For the experiments in Figs EV4 and EV5, cells were diluted to $3 \times 10^6$, $3 \times 10^5$, $3 \times 10^4$ and $3 \times 10^3$ cells/ml in PBS. Subsequently, spots containing $50 \times 10^4$, $50 \times 10^3$, $50 \times 10^2$ or 50 cells were then placed on the indicated media (YPD or YPD + 100 mM hydroxyurea), before growth for 2 days at 30°C.

### Measurement of DNA content

Cells were fixed and prepared for flow cytometry as described previously (Labib *et al*, 1999) and then analysed with a FACSCanto II flow cytometer (Becton Dickinson) and FlowJo software (Tree Star).

### Immunoprecipitation of proteins from yeast cell extracts

Frozen yeast pellets from 1 l cultures (except for Figs 1B and 6A, and Appendix Figs S1 and S6C, for which 250 ml culture was used) were prepared as described previously (Maric *et al*, 2014) in 100 mM Hepes-KOH pH 7.9, 100 mM KOAc, 10 mM Mg(OAc)$_2$, 2 mM EDTA, 2 mM β-glycerophosphate, 2 mM NaF, 1 mM DTT, 1% protease inhibitor cocktail (P8215, Sigma-Aldrich), 1× "Complete protease inhibitor Cocktail" (05056489001, Roche; a 25× stock was made by dissolving one tablet in 1 ml of water). For each sample, 2–3 g of frozen cells was ground in a SPEX SamplePrep 6780 Freezer/Mill. Chromosomal DNA was digested with Universal nuclease (123991963, Fisher; 1,600 U/ml, except for Figs 1 and 6, Appendix Figs S1 and S6 where 400 U/ml was used) for 30 min at 4°C, before two consecutive centrifugations at $25,000 \times g$ for 30 min and $100,000 \times g$ for 1 h. The resulting cell extract was added to magnetic beads (Dynabeads M-270 Epoxy, 14302D, Life Technologies) coupled to rabbit IgG (S1265, Sigma-Aldrich) and incubated for 2 h at 4°C. Proteins were separated on a 4–12% Bis-Tris NuPAGE gel (WG1402BOX, Life Technologies) in 1× MES and detected by immunoblotting, using previously characterized antibodies (Gambus *et al*, 2006, 2009; Foltman *et al*, 2013; Maric *et al*, 2014).

For mass spectrometry analysis, gels were stained with colloidal Coomassie (Simply Blue, LC6060, Thermo Fisher) and each sample lane was then cut into 40 bands before digestion with trypsin. Peptides were analysed by nanoliquid chromatography–tandem mass spectrometry with an Orbitrap Velos (MS Bioworks).

### Gene silencing assays at sub-telomeric, mating type or rDNA loci

The insertion of the *ADE2* marker near the right telomere of chromosome 5 was described previously (Iida & Araki, 2004; Foltman *et al*, 2013). We also generated an insertion of *ADE2* at the HMR locus on chromosome 3 (Foltman *et al*, 2013), but further analysis showed that the integration of *ADE2* was aberrant, as the marker did not segregate correctly in subsequent crosses (CE, unpublished data). This explains why we did not previously detect a defect in mating-type silencing in *mcm2-3A* cells (Foltman *et al*, 2013). In contrast, the *HMR::ADE2* and *HMR::2EDA* strains in this study were derived by crossing the previously described strains YLS409 (*HMR:: ADE2* with the *ADE2* promoter proximal to the *HMR-E* silencer element) and YLS410 (*HMR::2EDA* with the *ADE2* promoter distal to the *HMR-E* silencer element). The YLS409 and YLS410 strains (Sussel *et al*, 1993) were kindly provided by Professor David Shore (University of Geneva).

Diploid cells were generated that were heterozygous for the insertions of *ADE2* at the telomeric or mating-type loci. The cells were then sporulated and micro-dissected on YPD plates, using the Singer MSM system (Singer Instruments). The colonies were grown for 3 days at 30°C, followed by a further 3 days of incubation at 4°C to intensify the red colour associated with *ADE2* expression.

The rDNA silencing assays in Fig EV3 were based on strains generously provided by Danesh Moazed (Huang *et al*, 2006). Serial dilutions were made as described above, and then plated on "Synthetic Complete" medium (SC medium, non-selective for the *URA3* marker), or the equivalent mediums lacking Uracil (SC-Ura, selective for the *URA3* marker) for 2 days at 30°C, to detect expression of the "modified URA3" (mURA3) marker gene (Smith & Boeke, 1997) at the *leu2* locus (*leu2::mURA3*) or within the non-transcribed spacers 1 and 2 of an rDNA repeat (*NTS1::mURA3* and *NTS2::mURA3*).

## Telomere length assay

Yeast strains were grown in YPD medium until they reached 1.5–$2 \times 10^7$ cells/ml. Genomic DNA from each strain was isolated using the MasterPure Yeast DNA Purification Kit (MPY80200, Epicentre), and then, 6 µg was digested with XhoI for 12 h at 37°C. The digested DNA was resolved in a 0.8% agarose gel in 0.5× TBE for 24 h at 25 V, then transferred to a nitrocellulose membrane and hybridized with a probe specific to the Y' element of the telomere. The probe was labelled with $^{32}$P-dCTP using a Random Primed DNA labelling Kit (Roche 11 004 760 001). The membrane was exposed to BAS Imaging Plates (Fujifilm), which were then analysed using a Fujifilm FLA-5100 scanner and AIDA Image Analysis software (Raytest).

## Cloning and expression of human POLA1-NTD and MCM2-NTD

Synthetic gene fragments encoding the wild-type and mutated human POLA1 N-terminal domain (HsPOLA1-NTD) residues M1 to G110 and human MCM2 N-terminal domain (HsMCM2-NTD) residues G43 to D160, followed by a Twin-Strep tag, were purchased from Integrated DNA Technologies (IDT; Iowa, USA). The synthetic DNA fragments were designed with appropriate restriction sites at their 5′ and 3′ termini to allow insertion of HsPOLA1NTD-Twin-Strep into the expression vectors pOP5MP (for HsPOLA1NTD-Twin-Strep) and pOP3MP (for HsMCM2NTD-Twin-Strep), both of which introduce an N-terminal 8His-MBP tag that is cleavable by PreScission Protease (http://hyvonen.bioc.cam.ac.uk/pOP-vectors/); note that the C-terminal Avi-tag in pOPMP3 was removed during the cloning procedure. The amino acid sequences encoded by the resulting constructs are provided in Appendix Fig S8.

The expression plasmids were transformed into *E. coli* Rosetta2 (DE3) cells and grown in 1 l of 2× YT medium (Formedium; Hunstanton, UK) + 50 µg/ml Ampicillin (Melford; Chelsworth, UK) at 37°C to an $OD_{600}$ of about 0.4. The temperature was lowered to 20°C and after continued growth to an $OD_{600}$ of about 0.7, and protein expression was induced overnight by addition of 1 mM isopropyl-β-D-thiogalactopyranoside (Generon; Slough, UK). Cells were harvested by centrifugation at $4,000 \times g$ for 10 min and the pellet was resuspended in 25 ml buffer A [50 mM Tris–HCl pH 8.0, 500 mM NaCl, 2 mM imidazole + 1 tablet EDTA-free SIGMAFAST protease inhibitors per 100 ml (Sigma; St. Louis, USA)] and then snap-frozen in liquid nitrogen before storage at −80°C.

## Purification of HsPOLA1-NTD and HsMCM2-NTD

Cell pellets were thawed and 200 µl benzonase (25 U/µl, Merck; USA) were added. The cells were lysed by sonicating for 12 pulses of 10 s on and 20 s off, in iced water at 50% amplitude (Sonics VCX130; CT, USA). The lysed cells were centrifuged at $35,000 \times g$ for 45 min at 4°C, and the supernatant was then filtered (0.45 µm). The His-tagged proteins were purified by nickel affinity chromatography on a 1 ml gravity-flow Ni-NTA column (Qiagen; MD, USA). The column was equilibrated with 10 column volumes of equilibration buffer (20 mM Tris–HCl pH 8.0, 300 mM NaCl, 2 mM imidazole), and the filtered supernatant was loaded onto the column. The flow-through was discarded, and the column was then washed with 20 column volumes of equilibration buffer and 20 column volumes

of wash buffer (20 mM Tris–HCl pH 8.0, 300 mM NaCl, 10 mM imidazole). Proteins were then eluted with 10 column volumes of elution buffer (20 mM Tris–HCl pH 8.0, 300 mM NaCl, 300 mM imidazole), and 1 ml fractions were collected. Fractions containing the His-tagged protein were pooled, concentrated and loaded on a size-exclusion chromatography column (HiLoad 16/60 Superdex 200; GE Life-Science, USA) pre-equilibrated in SEC buffer (20 mM Tris–HCl pH 8, 300 mM NaCl). Peak fractions containing the purified proteins were pooled and incubated overnight at 4°C in the presence of GST-tagged PreScission Protease to remove His-MBP tandem tags. The digested proteins were then loaded onto a 1 ml GSTrap HP column (GE Life-Science; USA) to remove the GST-tagged PreScission Protease, followed by a 1 ml gravity-flow Ni-NTA column to remove the His-tagged MBP. The purity of proteins was determined by SDS polyacrylamide gel electrophoresis and protein concentration was determined by UV spectroscopy, using theoretical $\varepsilon_{280}$ (http://web.expasy.org/protparam/), and amino acid analysis (http://www3.bioc.cam.ac.uk/pnac/aaa.html).

## Preparation of high-salt HeLa nuclear extracts

The procedure for preparing HeLa cell extracts was adapted from a previously described protocol (Abmayr *et al*, 2006). All operations were performed at 0–4°C. Initially, $5 \times 10^9$ HeLa cells (IpraCell; Mons, Belgium) were washed in 50 ml PBS buffer, and the cell pellet was then resuspended in five packed cell volumes of lysis buffer (10 mM Tris–HCl pH 8.0, 100 mM NaCl, 0.15 mM spermine, 0.5 mM spermidine, 0.5 M sucrose, 2 mM EDTA, 0.1% Igepal, 2 mM DTT and 1 tablet SIGMAFAST protease inhibitors per 100 ml). Cells were homogenized with a glass Dounce homogenizer with 15 up-and-down strokes using a type B pestle. Nuclei were then collected by centrifuging at $3,000 \times g$ for 10 min and were resuspended in one packed nuclear volume of low-salt buffer (20 mM HEPES-NaOH pH 7.8, 25% glycerol, 1.5 mM $MgCl_2$, 20 mM KCl, 1 mM EDTA, 2 mM DTT, one tablet SIGMAFAST protease inhibitors per 50 ml). Following incubation for 10 min on rollers, the sample was spun at $5,000 \times g$ for 10 min and the nuclear pellet was slowly resuspended in two packed-nuclear-volumes of high-salt buffer 1 (20 mM HEPES-NaOH pH 7.8, 25% glycerol, 1.5 mM $MgCl_2$, 1 M KCl, 1 mM EDTA, 2 mM DTT and one tablet SIGMA-FAST protease inhibitors per 50 ml) and then homogenized with 10 up-and-down strokes. Nuclei were incubated for 30 min with continuous rotation and 200 mg of polyethyleneimine (Polysciences; IL, USA) was added. The nuclear debris and DNA were pelleted at $50,000 \times g$ for 30 min, and the supernatant was collected. 50 mg of polyethyleneimine was added to the supernatant that, after a 2-h incubation on rollers, was then dialysed against high-salt buffer 2 (20 mM HEPES-NaOH pH 7.8, 25% glycerol, 1.5 mM $MgCl_2$, 1 M NaCl, 1 mM EDTA, 2 mM DTT) for 2 h. Following centrifugation at $50,000 \times g$ for 30 min, the supernatant, corresponding to the high-salt nuclear extract, was collected, aliquoted and stored at −80°C.

## Pull-down assays for HsPOLA1-NTD and HsMCM2-NTD

The purified HsPOLA1-NTD and HsMCM2-NTD fragments were bound to beads via the Twin-Strep tag at the C-terminal end of each protein. All operations were performed at 0–4°C, using the

"pull-down buffer" (PD buffer) containing 20 mM Tris–HCl pH 8.0, 150 mM NaCl, 1 mM DTT, 5% glycerol and 0.005% Tween-20 plus one tablet SIGMAFAST protease inhibitors per 50 ml buffer. For each pull-down assay, 300 μl Strep-Tactin Superflow 50% suspension resin (IBA; Goettingen, Germany) was washed twice in 1 ml PD buffer, before addition of 45 μl of Twin-Strep-tagged bait at 40 μM concentration, together with 150 μl PD buffer. Following a 1-h incubation on rollers, the samples were washed twice with 1 ml PD buffer. 150 μl High-salt HeLa nuclear extract was then added to the resin, together with 150 μl PD buffer, and after a 1-h incubation on rollers, the samples were washed twice with 1 ml PD buffer. Elution was performed by adding 150 μl elution buffer (7.5 mM d-Desthiobiotin (Sigma) dissolved in PD buffer) to the pelleted resin. Following a 15-min incubation, the resin was spun down at 5,000 g for 3 min and 90 μl of the supernatant was acid precipitated by the addition of 30 μl, ice-cold, trichloroacetic acid and incubation for 30 min on ice. The precipitated proteins were pelleted at 20,000 g for 10 min, the supernatant was aspirated off and the pellets washed once with 1 ml acidified acetone (10 mM HCl in acetone) and twice in pure acetone. The pellets were air-dried overnight, resuspended in denaturing gel loading buffer containing fresh 8 M urea and analysed on Bis-Tris 20% acrylamide gels (29:1 Acrylamide : Bis-Acrylamide). Gel bands for the pulled-down proteins were identified by MALDI fingerprinting (http://www3.bioc.cam.ac.uk/pnac/proteomics.html).

### Cloning and expression of human histones H2A and H2B

Synthetic genes coding for human histones H2A type 1 (UniProt P0C0S8) and human H2B type 1-A (UniProt Q96A08) were purchased from Integrated DNA Technologies (IDT; Iowa, USA). The synthetic genes were codon-optimized for expression in *E. coli*, with appropriate restriction sites at their 5′ and 3′ termini to allow insertion into the dual expression vector pRSFDuet (Novagen, Merck; Darmstadt, Germany). The synthetic H2A gene was inserted into the Multiple Cloning Site 1 (MCS1) downstream from the vector hexa-histidine site, followed by a PreScission protease site. The synthetic H2B gene was cloned into MCS2.

The expression plasmid was transformed into *E. coli* Rosetta2 (DE3) cells, and H2A-H2B expression was induced as described above for HsPOLA1-NTD and HsMCM2-NTD. The harvested cell pellet was resuspended in 25 ml (per 1 l cell culture) of lysis buffer (50 mM sodium phosphate pH 7.8, 2 M NaCl, 20 mM imidazole), then snap-frozen in liquid nitrogen and stored at −80°C.

### Purification of human H2A-H2B dimer

The thawed cell pellet was lysed by sonicating at 50% amplitude (Sonics VCX130; CT, USA) for 12 pulses of 20 s on and 40 s off, in iced water. The lysed cell extract was centrifuged at 35,000 × g for 45 min at 4°C, and the supernatant was then filtered (0.45 μm). The His-tagged proteins were purified by nickel affinity chromatography on a 1 ml HisTrap column (GE Life-Science, USA). The column was equilibrated with five column volumes of lysis buffer, and the filtered supernatant was loaded onto the column. The flow-through was discarded, and the column was washed with 20 column volumes of lysis buffer. The H2A-H2B dimer was eluted with 50 mM sodium phosphate pH 7.8, 2 M NaCl, 300 mM imidazole,

and 1 ml fractions were collected. Fractions containing the His-tagged H2A-H2B dimer were pooled, concentrated and loaded on a size-exclusion chromatography column (HiLoad 16/60 Superdex 75; GE Life-Science, USA) pre-equilibrated in 20 mM Tris–HCl pH 8.0, 2 M NaCl, 0.1 mM TCEP). The peak fractions containing the purified dimer were pooled and concentrated in an Amicon Ultra-15 spin column (3000 MWCO; Merck; Darmstadt, Germany) at 4°C to a volume of about 2 ml (≅ 100 μM H2A-H2B dimer). The NaCl concentration was reduced from 2 M to 0.25 M by twofold dilution steps with 20 mM Tris–HCl pH 8.0, 1 mM DTT and the sample was then incubated overnight at 4°C in the presence of GST-tagged PreScission Protease, in order to remove the hexa-histidine tag. The digested dimer was loaded onto a 1 ml GSTrap HP column (GE Life-Science; USA) to remove the GST-tagged PreScission protease, and the flow-through was then loaded onto a 1 ml HiTrap-Heparin column (GE Life-Science, USA). The column was equilibrated with 20 mM Tris–HCl pH 8.0, 100 mM NaCl, 0.1 mM TCEP, and H2A-H2B dimer was eluted with a 20-column volume gradient that ranged from the above "low-salt" buffer to 20 mM Tris–HCl pH 8.0, 2 M NaCl, 0.1 mM TCEP "high-salt" buffer. Fractions containing pure H2A-H2B dimer were pooled, concentrated, snap-frozen in liquid nitrogen and stored at −80°C. The purity of the proteins was determined by SDS polyacrylamide gel electrophoresis and the protein concentration was determined by UV spectroscopy, using the theoretical $\varepsilon_{280}$ value (http://web.expasy.org/protparam/). The integrity of the human H2A-H2B dimer was determined by multi-angle light scattering (MALS) in 20 mM Tris–HCl pH 8.0, 150 mM NaCl, 5% glycerol, 0.005% Tween 20 (not shown).

### Isothermal titration calorimetry

Isothermal titration calorimetry experiments were conducted using a MicroCal iTC200 instrument (Malvern Panalytical; Malvern, UK). Prior to the experiments, human H2A-H2B dimer, wild-type HsPOLA1-NTD and HsPOLA1-NTD-6A were re-purified on a Superdex 75 10/300 GL column (GE Life-Science, USA) equilibrated in ITC buffer (20 mM Tris–HCl pH 8.0, 150 mM NaCl, 5% glycerol, 0.005% Tween 20). Titration experiments were carried out at 25°C using 19 injections and 750 r.p.m. The first injection was 0.4 μl at 0.5 μl/s and was discarded from the data analysis. All the following injections were 2 μl at 0.5 μl/s with 180-s spacing between each injection. 80 μM Pol1-NTD fragments were injected into 10 μM H2A-H2B dimer in the cell. For controls, 80 μM Pol1-NTD constructs were injected into ITC buffer and the heat evolved was subtracted from the heat produced by the interaction of HsPol1NTD with the H2A-H2B dimer. The data were analysed using a single-site binding model and a non-linear regression analysis using ORIGIN software (OriginLab Corporation, USA). The variables for ΔH (enthalpy change), N (stoichiometry) and $K_a$ (association equilibrium constant) were allowed to "float" during the analysis.

**Expanded View** for this article is available online.

## Acknowledgements

We gratefully acknowledge the support of the Wellcome Trust (references 102943/Z/13/Z and 104641/Z/14/Z for investigator awards to KL and LP respectively, plus reference 097945/B/11/Z for flow cytometry), and the Medical Research Council (core grant MC_UU_12016/13). We thank Magda Foltman for

assistance in the early stages of this project; Arturo Calzada for the telomeric DNA probe; Hiroyuki Araki, Danesh Moazed and David Shore for strains; Fabrizio Villa for help with running CHEF gels; and MRC PPU Reagents and Services (https://mrcppureagents.dundee.ac.uk) for plasmids and antibodies.

## Author contributions

LP noticed the similarity between the amino terminus of Pol1 and the histone-binding motif of Mcm2. KL and CE conceived and designed the project and CE performed all experiments, except for those in Fig 5 and Appendix Fig S8 that were carried out by JDM and AD. KL wrote the manuscript with contributions and critical comments from CE, JM and LP.

## Conflict of interest

The authors declare that they have no conflict of interest.

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
