## [Review Process File · The EMBO Journal]

Histone H2A-H2B binding by Pol a in the eukaryotic replisome contributes to the maintenance of repressive chromatin

Cecile Evrin, Joseph D. Maman, Aurora Diamante, Luca Pellegrini and Karim Labib.

Review timeline:

Submission date:	12 th January 2018
Editorial Decision:	5 th February 2018
Revision received:	18 th June 2018
Accepted:	24 th July 2018

Editor: Hartmut Vordermaier

Transaction Report:

1st Editorial Decision

5th February 2018

Thank you for submitting your manuscript on polymerase alpha tethering of released H2A/H2B dimers for our editorial consideration. We have now received the feedback from three expert reviewers, copied below for your information. As you will see, all referees find this work interesting and potentially important, but the reports also raise a number of well-taken queries and specific experimental concerns requiring clarification. Should you be able to satisfactorily address these various issues, we shall be happy to consider a revised version of this study further for publication in The EMBO Journal. Since our policy to allow only a single round of major revision makes it important to carefully answer to all points raised at this stage, please do not hesitate to get back in case you would like to discuss specific referee request or should need guidance on further-reaching experiments. We might also discuss possible extension of the revision period (beyond the regular three months), during which time the publication of any competing work elsewhere would have no negative impact on our final assessment of your own study.

REFeree REPORTS.

Referee #1:

Evin et al. described a histone binding motif in Pol α and found this motif mediates the Pol α -H2A/H2B interaction. They also found that the histone binding activity of Pol α is dispensable for Pol α 's function during DNA replication. Moreover, they claimed that this Pol α -H2A/H2B interaction contributes to histone processing by the eukaryotic replisome, thereby contributing to sub-telomeric gene silencing. Overall, while I found that the discovery of the histone binding motif in Pol α is interesting, I feel that many aspects of experimental results need be strengthened to support their conclusions.

Major points:

1, The authors performed pull down assay using HeLa cell extracts to demonstrate that POLA1-NTD binds histone H2A/H2B. As IP experiments from yeast extracts showed that all four histones, including H2A, H2B, H3 and H4, present in Pol1 associated protein complex, it is very important to know whether Pol1 binds H2A/H2B or H3/H4 or both. Therefore, I suggest the authors using recombinant histone octamers to test whether Pol1 binds to histones using Pol1-NT and Pol1-full length in in vitro pull down assays. Moreover, the effect of Pol1-3A on histone binding in vitro is not as dramatic as in yeast cells. Therefore, the Pol-6A mutant should be used as controls in these experiments. It would also be better to include salt titrations and protein amounts titrations in these experiments.

2, The authors are trying to establish a new function of Pol1 in eukaryotic replisome, however, the majority of the IP experiments were performed in G2-M cell extracts, and I wondered how this condition could reflect the real situation where eukaryotic replisome performs its function in S phase. The authors should test Pol1-histone interactions of the WT and mutant forms during S phase. In addition, the authors should construct at least one Pol1 mutant such as Pol1-6A at endogenous chromosome locus to test the interaction of Pol1-histones in this strain.

3, In Figure 3E, the authors employed a *spt16-ΔCT* to test the interaction of Pol1-NT with FACT while Spt16 could not bind H3/H4, all four histone signals in the IP-associated complex should be shown in this figure. It would be also useful to perform the same experiment in S phase.

4. The telomeric ADE2 silencing assay is a very sensitive assay for telomeric silencing. The authors should test whether the silencing phenotype could be seen in other heterochromatin regions such as HM loci.

5, It would be interesting to go further how Pol α binding with H2A-H2B contribute to telomere silencing. The author proposed that for recycle parent H2A/H2B at forks therefore impacts on epigenetic states. However, this conclusion does not agree what people know about H2A/H2B deposition during S phase. It is known that nucleosomal H2A/H2B exchanged "freely" with cytosolic H2A/H2B during one cell cycle. Therefore, it is important to monitor the impact of the effect of Pol1 mutation on the deposition of parental H2A/H2B in yeast cells during S phase using ChIP based assays.

6, DNA replication and histone dynamics during S phase is a highly coordinated process. It was established in mammalian cells the histone demand and supply affect the movement of replication fork. It is hard to imagine that if the Pol α involved in parental histone process is important during S phase will led to no impact on DNA replication. Combining with *mec1Δ* shows little growth defect doesn't necessary reflect that this one doesn't contribute to DNA replication, I would suggest that the author modify this part.

Minor points:

1, Figure 3F, it would be better if the author could also show H2A or H2B in the same experiments

2. Figure 4D, The bands in Hs WT IP are too faint to be seen, it would be better if the author could show a repeat results with a higher quality.

3. It would be more convincing to present a FACS result in 10 min-intervals to monitor the S phase progression in Pol1 mutant cells.

4, It was reported that *S. pombe* Pol α has a role in epigenetic silencing. The author should cite and discuss this reference "A role for DNA polymerase α in epigenetic control of transcriptional silencing in fission yeast" (EMBO J, 2001, V20: 2857-2866)

Referee #2:

This manuscript describes experiments that investigate the role of DNA polymerase alpha in eukaryotic DNA replication beyond its role in synthesis of Okazaki fragments. The authors describe studies demonstrating that the catalytic subunit of Pol alpha contains a motif at its N-terminus whose role is independent of DNA synthesis, but rather facilitates the re-loading of histones H2A and H2B after replication, thus controlling the process of sub-telomeric gene silencing, and, along with FACT and Mcm2, helping to conserve chromatic states during nuclear DNA replication.

Comments: The experiments presented here are convincing, the discussion is thorough and interesting, and the manuscript is beautifully written. I believe it is worthy of publication as is. That said, I do wonder about the point mutation rate of the pol1-2A2 and/or pol1-6A cells. Could their effects on histone processing have consequence for mismatch repair following replication? If so, such an effect might link these two fields of study.

Referee #3:

Labib and colleagues describe an investigation of the role of Ctf4-dependent DNA Pol a tethering to the replisome. They find that this tethering is dispensable for normal DNA replication, consistent with previous studies showing that Ctf4 is not required for replication *in vivo* or *in vitro*. Instead the authors find that Ctf4-dependent tethering is involved in maintaining a heterochromatic state at the telomeres of yeast. They go on to find that maintenance of this state requires a histone-binding domain in the N-terminus of Pol a. Interestingly, this motif is also required for the association of Pol a with the Mcm2 and FACT and these associations appear to be mediated by interaction with the same set of histones. The authors identify a histone binding motif on Pol a and show that it is conserved and that the human variant binds histone H2A and H2B in contrast to the histone-binding motif in human MCM2 that binds H3 and H4 (providing a nice explanation of how these proteins simultaneously bind to a histone assembly - either a nucleosome or hexasome). Based on these data, the authors suggest that these motifs act together to retain displaced histones in the same location on newly replicated DNA.

The conclusions of this paper are well supported by the data and it is appropriate for publication. That the mutations interfering with the histone binding motifs only impact chromatin at the telomeres seems unlikely and it would be interesting to look at genome-wide gene expression to see if additional sites could be identified. That being said, it would be best to do this in an organism with a more complex chromatin states to inherit (e.g. mammalian cells) and that is beyond the scope of the current study. One oddity of the data is that the modifications responsible for regulation of telomere-proximal gene expression involve modifications of H3 and H4 rather than H2A and H2B. Thus, it is surprising that a chaperone of H2A/H2B would regulate this event. This suggests that it is complex bigger than the H3/H4 tetrasome that is retained. The authors indirectly address this on page 19 but they should connect this discussion to the types of modifications that mediate telomeric silencing to further support the idea of a hexasome being passed from in front to behind the replication fork. This is a very interesting implication of the study.

1st Revision - authors' response

18th June 2018

We thank all the reviewers for their helpful comments and their interest in our work. The revised manuscript contains extensive new data as discussed below, together with textual changes to address the various points that were raised.

Referee #1

The reviewer summarized his/her view by saying “*while I found that the discovery of the histone binding motif in Pola is interesting, I feel that many aspects of experimental results need be strengthened to support their conclusions.*”

Major points:

1, “*The authors performed pull down assay using HeLa cell extracts to demonstrate that POLA1-NTD binds histone H2A/H2B. As IP experiments from yeast extracts showed that all four*

histones, including H2A, H2B, H3 and H4, present in Pol1 associated protein complex, it is very important to know whether Pol1 binds H2A/H2B or H3/H4 or both.”

It is very important to note that the yeast IP data do not conflict with the HeLa pulldown data – the two assays are very different and they serve distinct purposes in our study.

Previous studies showed that HeLa cell extracts contain ‘free’ H3-H4 and H2A-H2B histone sub-complexes that are available to bind to their corresponding chaperones (e.g. previous work from Anja Groth showing that ASF1 and MCM2 are jointly able to bind to H3-H4 dimers in HeLa extracts). HeLa extracts are thus a useful assay system with which to determine which histone sub-complexes (H2A-H2B or H3-H4) can bind to which chaperones, and we used this system to show that human MCM2 NTD binds preferentially to H3-H4 (as shown previously) whereas human POLA1 NTD binds preferentially to H2A-H2B.

In contrast, the yeast IP experiments monitor the ability of factors to bind to histone-complexes that have been released from chromatin by DNase treatment. These complexes contain H3-H4 tetramers associated with H2A-H2B dimers, and so the assay does not distinguish which factors bind to which histones. We previously showed (Foltman et al, 2013) that the chaperones that build new nucleosomes at replication forks (e.g. Asf1) are negative in this assay, since such chaperones only bind to histone surfaces that are subsequently hidden within the nucleosomal histone octamer. In contrast, Mcm2 NTD, FACT and Pol1 NTD are all able to bind to chromatin-derived histone complexes containing all four histones, likely in the form of hexamers. Moreover, we find that Mcm2 NTD, FACT and Pol1 NTD can bind simultaneously to the same histone complexes (and so must bind to distinct surfaces).

The point of the yeast IP assay is that it reflects the kind of histone-binding activity that is required at replication forks (the ability to bind chromatin-derived histone complexes that contain H3-H4 tetramers). We note that HsPOLA1-NTD and ScPol1-NTD behave similarly in this assay (though binding of HsPOLA1-NTD to the yeast histone complexes is understandably weaker). Moreover, histone-binding by both the human and yeast Pol alpha tails is abrogated to a similar degree in this assay by the same ‘2A2’ and ‘6A’ mutations (Figure 4).

“Therefore, I suggest the authors using recombinant histone octamers to test whether Pol1 binds to histones using Pol1-NT and Pol1-full length in in vitro pull down assays. Moreover, the effect of Pol1-3A on histone binding in vitro is not as dramatic as in yeast cells. Therefore, the Pol-6A mutant should be used as controls in these experiments. It would also be better to include salt titrations and protein amounts titrations in these experiments.”

Our new data in Figure 5B now show by isothermal titration calorimetry that HsPOLA1-NTD binds *in vitro* to recombinant histone H2A-H2B dimer with 1:1 stoichiometry and a K_d of 19 nM. Moreover, Figure 5C shows that the HsPOLA1-6A mutant has no detectable binding in the ITC assay.

We plan to analyse the binding of yeast Pol1-NTD to recombinant yeast histones in our future experiments, but thus far this has not been possible for technical reasons. Our new data in Appendix Figure S3 shows that the binding of yeast Pol1-NTD to chromatin-derived yeast histone complexes is very salt sensitive, which makes the recombinant assays more difficult (recombinant histones prefer high salt). Our preliminary findings support our observations with human POLA1-NTD (please see ‘Figure for reviewers only’, which shows that yeast Pol1-NTD interacts in ITC experiments with human histones H2A-H2B, with a K_d of 1 μM under these conditions), but this issue will need future work that is beyond the scope of the present study.

2. “The authors are trying to establish a new function of Pol1 in eukaryotic replisome, however, the majority of the IP experiments were performed in G2-M cell extracts, and I wondered how this condition could reflect the real situation where eukaryotic replisome performs its function in S phase. The authors should test Pol1-histone interactions of the WT and mutant forms during S phase.”

We have now performed the suggested experiments with S-phase extracts, and the new data are contained in Appendix Figure S5A-B, showing that Pol1NT binds chromatin-derived histone complexes similarly in S-phase or G2-M phase extracts, and in both cases the binding is lost for the 6A mutant.

As noted above, the IP assay monitors the ability of individual replisome components to bind to chromatin-derived histone complexes that contain all four histones, since this will be the relevant activity at replication forks, and there is no reason to expect that chromatin-derived histones from S-phase or G2-M phase will behave differently in such an assay.

We used G2-M extracts, as in our previous study (Foltman et al, 2013), since the various replisome components all associate with each other to form the replisome in S-phase. This

complicates the analysis, since it is harder to distinguish which factors are binding to histones, and which factors are simply binding to other replisome components (that might bind to histones). In G2-M phase, the replisome has fallen apart into its component parts, so the assay is simpler to interpret.

“In addition, the authors should construct at least one Pol1 mutant such as Pol1-6A at endogenous chromosome locus to test the interaction of Pol1-histones in this strain”.

We have performed the requested experiment with S-phase extracts, and the new data are contained in Appendix Figure S6A, showing that the co-purification of chromatin-derived histones with Pol1-6A is reduced in comparison to wild type Pol1 (as noted above, the interpretation of the residual histone-binding in the mutant is complicated by the fact that Pol1 associates with the replisome in S-phase, and is thus bound to other factors that also have histone-binding activity).

3. “In Figure 3E, the authors employed a spt16-ΔCT to test the interaction of Pol1-NT with FACT while Spt16 could not bind H3/H4, all four histone signals in the IP-associated complex should be shown in this figure. It would be also useful to perform the same experiment in S phase.”

We appreciate the reviewer’s point – the requested data are now provided in Appendix Figure S4. The experiment in Figure 3E involved an IP of Pol1NT from an extract of cells that contained endogenous wild type FACT, in addition to a second copy of tagged wild type or mutant Spt16. So in fact we would see all four histones in the IP regardless of the status of the tagged second copy of Spt16 (wt or mutant). For this reason, we omitted the histones in Figure 3E for the sake of simplicity (the point of the experiment was simply to show that co-purification of Pol1NT with Spt16 requires histone-binding activity of the latter).

The complete version of the experiment in Appendix Figure S4 now shows all four histones as requested (panel A), together with an IP of tagged Spt16 to show that the mutant version did not co-purify with histones (panel B). Moreover, panels C-D present an equivalent experiment to the one in Figure 3E, but using S-phase extracts as requested.

4. “The telomeric ADE2 silencing assay is a very sensitive assay for telomeric silencing. The authors should test whether the silencing phenotype could be seen in other heterochromatin regions such as HM loci.”

We are grateful to the reviewer for this helpful suggestion. We now present new data in Figure 2D-F, Figure 7C-D and Figure EV2, showing that mating-type silencing at HMR is indeed defective in *pol1-4A* (Pol alpha displaced from replisome) as well as in *pol1-2A2* and *pol1-6A* (histone-binding mutants of Pol1). These data show that the replisome-binding and histone-binding activities of Pol alpha are required for both of the classes of silent chromatin in yeast (mating-type and telomeric) that are dependent upon the Sir2-Sir3-Sir4 proteins. This presumably reflects a common underlying mechanism, which we discuss on pages 20-23.

The replisome is not required to preserve rDNA silencing, which is independent of Sir3-Sir4 and involves more than one mechanism. In the histone-binding mutants, rDNA silencing can presumably be re-established between completion of one round of S-phase and initiation of the next. This issue is discussed on pages 21-22.

5. “It would be interesting to go further how Pola binding with H2A-H2B contribute to telomere silencing. The author proposed that for recycle parent H2A/H2B at forks therefore impacts on epigenetic states. However, this conclusion does not agree what people know about H2A/H2B deposition during S phase. It is known that nucleosomal H2A/H2B exchanged “freely” with cytosolic H2A/H2B during one cell cycle. Therefore, it is important to monitor the impact of the effect of Pol1 mutation on the deposition of parental H2A/H2B in yeast cells during S phase using ChIP based assays.”

The key points are that:

- our data show that Pol1NT binds to chromatin-derived histone complexes together with FACT and Mcm2 (they can all bind simultaneously to the same complexes)
- histone-binding mutations in Pol1 and Mcm2 share the same phenotypes.

These data do not disagree with previous observations that nucleosomal H2A-H2B exchange freely. Instead, our data relate to one specific function of the replisome – namely the role of the replisome in the transfer of parental histone complexes (containing both H3-H4 and H2A-H2B) at replication forks.

As noted by reviewer 3, our data indicate that the unit of transfer of histones at replication forks is bigger than a simple H3-H4 tetramer (it’s likely to be a hexamer). However, after the initial

transfer of parental histones onto nascent DNA, histones H2A-H2B will then exchange rapidly, as described previously (we discuss this point on pages 23, lines 480-483).

So we do not think that the suggested ChIP experiments would provide interpretable data, since H2A-H2B ChIP data should look similar in wt and mutant cells, given the rapid exchange of H2A-H2B.

Our data do relate to the fate of H2A-H2B after passage of a replication fork (they will exchange rapidly), but instead relate specifically to the mechanism by which parental histones are transferred after release from DNA at replication forks (our data indicate that the initial unit of transfer involves a complex that contains both H3-H4 and H2A-H2B). Though Pol1NT binds H2A-H2B, it is important to preserve a silencing phenomenon that is dependent on preserving de-acetylation of H3-H4. Our data suggest that a mechanism for this phenotype, namely that Pol1NT acts with Mcm2 and FACT to transfer parental histone complexes (containing all four histones) onto nascent DNA at replication forks.

6. “DNA replication and histone dynamics during S phase is a highly coordinated process. It was established in mammalian cells the histone demand and supply affect the movement of replication fork. It is hard to imagine that if the Pola involved in parental histone process is important during S phase will led to no impact on DNA replication. Combining with mec1Δ shows little growth defect doesn't necessary reflect that this one doesn't contribute to DNA replication, I would suggest that the author modify this part.”

As noted above, the histone-binding motifs of Pol1 and Mcm2 relate to one specific function of the replisome – namely the mechanism of transfer of parental histones at replication forks – and do not relate to ‘histone demand and supply’ (assuming that this means the provision of newly synthesized histones to generate new nucleosomes).

Our data indicate that DNA synthesis is unaffected by mutation of the CIP-box or the histone binding motif of Pol1, since these mutations do not affect telomere length (a sensitive measure of Pol alpha-primase function), do not affect replisome assembly or DNA synthesis progression, and do not lead to synthetic lethality with *mec1Δ*. Moreover, our data agree with recent *in vitro* data from John Diffley’s lab (Yeeles et al, Mol Cell, 2017), which showed that Ctf4 (and thus the replisome-tethering of Pol alpha) is dispensable for efficient DNA replication. Therefore, the simplest interpretation of our data is indeed that DNA synthesis per se is not affected by the *pol1-4A*, *pol1-2A*, *pol1-2A2* or *pol1-6A* mutations, which instead cause a specific defect in Sir2-3-4-dependent gene silencing.

Minor points:

1. “Figure 3F, it would be better if the author could also show H2A or H2B in the same experiments”

The revised version of Figure 3F shows all four histones.

2. “Figure 4D, The bands in Hs WT IP are too faint to be seen, it would be better if the author could show a repeat results with a higher quality.”

The revised version of Figure 4D is of improved quality.

3. “It would be more convincing to present a FACS result in 10 min-intervals to monitor the S phase progression in Pol1 mutant cells.”

New data in Figure 1A, Figure 6B and Appendix Figure S6D address this point.

4. “It was reported that *S. pombe* Pol α has a role in epigenetic silencing. The author should cite and discuss this reference “A role for DNA polymerase α in epigenetic control of transcriptional silencing in fission yeast” (EMBO J, 2001, V20: 2857-2866)”

We thank the reviewer for pointing this out, and we have included discussion and citation of this paper on page 25, lines 518-522.

Referee #2:

The reviewer summarized his/her view by saying:

“The experiments presented here are convincing, the discussion is thorough and interesting, and the manuscript is beautifully written. I believe it is worthy of publication as is.”

The reviewer went on to say:

“That said, I do wonder about the point mutation rate of the *pol1-2A2* and/or *pol1-6A* cells. Could their effects on histone processing have consequence for mismatch repair following replication?”

If so, such an effect might link these two fields of study.”

We agree that it would be interesting if future studies were to establish a link between histone processing and mismatch repair, but unfortunately we did not have had time to address this point within the timeframe of the revised manuscript.

However, we hope that the reviewer might appreciate the additional data that we have added, particularly the demonstration that gene silencing in the HMR mating type locus is defective in the *poll-4A* mutant (Pol alpha displaced from the replisome) and the *poll-2A2* and *poll-6A* mutants (mutations in the novel histone-binding motif of Pol1), extending our earlier observations regarding telomeric silencing, and showing that the replisome's histone binding activities are required to preserve gene silencing that is dependent on Sir2-3-4 (in contrast to rDNA silencing that involves other mechanisms).

Referee #3:

The reviewer summarized his/her view by saying:

“The conclusions of this paper are well supported by the data and it is appropriate for publication. That the mutations interfering with the histone binding motifs only impact chromatin at the telomeres seems unlikely and it would be interesting to look at genome-wide gene expression to see if additional sites could be identified. That being said, it would be best to do this in an organism with a more complex chromatin states to inherit (e.g. mammalian cells) and that is beyond the scope of the current study.”

In our revised manuscript, we present new data showing that mating type silencing is defective in the *poll-4A* mutant (Pol alpha displaced from the replisome) and the *poll-2A2* and *poll-6A* mutants (mutations in the novel histone-binding motif of Pol1), in addition to the defect in telomeric gene silencing.

We agree that it will be interesting in the future to examine the consequence of mutating the replisome's histone-binding activities in an organism with more complex chromatin states.

“One oddity of the data is that the modifications responsible for regulation of telomere-proximal gene expression involve modifications of H3 and H4 rather than H2A and H2B. Thus, it is surprising that a chaperone of H2A/H2B would regulate this event. This suggests that it is complex bigger than the H3/H4 tetrasome that is retained. The authors indirectly address this on page 19 but they should connect this discussion to the types of modifications that mediate telomeric silencing to further support the idea of a hexasome being passed from in front to behind the replication fork. This is a very interesting implication of the study.”

We thank the reviewer for this helpful suggestion, and we have now extended and re-written the corresponding section of the Discussion (pages 20-23).

Accepted

24th July 2018

Thank you for submitting your revised manuscript for our consideration. It has now been seen once more by two of the original referees (see comments below), and I am happy to inform you that there are no further objections towards publication in The EMBO Journal.

Referee #1 (Report for Author)

The authors have significantly improved the manuscript. I support the acceptance of this manuscript for publication.

Referee #3 (Report for Author)

The revised manuscript improves upon an already excellent study. The addition of the mating-type silencing data is particularly nice. I fully support publication.

Corresponding Author Name: KARIM LABIB

Journal Submitted to: THE EMBO JOURNAL

Manuscript Number: EMBOJ-2018-99021